# The cryo-EM structure of the bacterial flagellum cap complex suggests a molecular mechanism for filament elongation

Natalie S. Al-Otaibi [1], Aidan J. Taylor[1], Daniel P. Farrell[2], Svetomir B. Tzokov [1], Frank DiMaio[2], David J. Kelly [1✉] & Julien R. C. Bergeron [1,3✉]

The bacterial flagellum is a remarkable molecular motor, whose primary function in bacteria is to facilitate motility through the rotation of a filament protruding from the bacterial cell. A cap complex, consisting of an oligomer of the protein FliD, is localized at the tip of the flagellum, and is essential for filament assembly, as well as adherence to surfaces in some bacteria. However, the structure of the intact cap complex, and the molecular basis for its interaction with the filament, remains elusive. Here we report the cryo-EM structure of the *Campylobacter jejuni* cap complex, which reveals that FliD is pentameric, with the N-terminal region of the protomer forming an extensive set of contacts across several subunits, that contribute to FliD oligomerization. We also demonstrate that the native *C. jejuni* flagellum filament is 11-stranded, contrary to a previously published cryo-EM structure, and propose a molecular model for the filament-cap interaction.

[1] Department of Molecular Biology and Biotechnology, the University of Sheffield, Sheffield, UK. [2] Department of Biochemistry, University of Washington, Seattle, WA, USA. [3]Present address: Randall Division of Cell and Molecular Biophysics, King's College London, London, UK. ✉email: d.kelly@sheffield.ac.uk; julien.bergeron@kcl.ac.uk

The bacterial flagellum is a macromolecular motor that rotates and acts as a propeller in many bacteria. It is associated with virulence in many human pathogens including *Salmonella*, enteropathogenic *Escherichia coli*, *Campylobacter*, and *Helicobacter* species[1,2]. The flagellum is composed of >25 different proteins, and consists of three main regions: the basal body acts as an anchor in the bacterial membrane, and includes the molecular machinery for rotation and protein secretion; the hook forms a junction which protrudes from the outer membrane; and the filament, consisting of multiple repeats of a single protein (flagellin), forms the propeller[3]. The filament, that can be >20 μm in length, is topped by a cap complex, that consists of several copies of the protein FliD. This complex initially attaches to the hook-filament junction, and using a yet unknown mechanism, assists in building the filament[4].

Low-resolution cryo-EM studies of the cap complex in *Salmonella enterica* have suggested that it consists of five copies of FliD (also known as HAP2), forming a "stool"-shaped complex with a core "head" domain and five flexible "leg" domains, that interact with the growing end of the filament[5,6]. Crystal structures of the FliD head domain have been reported for several species, and revealed a range of crystallographic symmetries, from tetramers in *Serratia marscecens* (FliD$_{sm}$), pentamers in *S. enterica* (FliD$_{se}$) to hexamers in *E. coli* (FliD$_{ec}$) and *Pseudomonas aeruginosa* (FliD$_{pa}$)[7–10]. This observation led to the hypothesis that the cap complex can have different, species-specific oligomeric states.

The flagellar filament has been studied extensively by cryo-EM, and its high-resolution structure has been reported in a range of bacteria, including *Bacillus subtilis*, *P. aeruginosa,* and *S. enterica*. In all of these, the filament was shown to consist of 11 protofilaments[11,12]. However, a low-resolution cryo-EM study of the *C. jejuni* flagellar filament suggested the presence of 7 protofilaments[13]. Taken together with the range of oligomeric states observed in the FliD crystal structures, these observations have led to a model where in different bacterial species, the cap complex has different oligomeric states (N), and in the corresponding filament, the number of protofilaments is $2N + 1$[7].

*Campylobacter jejuni* is a Gram-negative, spiral-shaped microaerophilic epsilon proteobacterium, colonizing the lower gastrointestinal (GI) tract of humans and poultry[14]. It is the most common cause of bacterial gastroenteritis and can lead to severe sequelae such as Guillain-Barré (GBS) and Miller-Fisher syndromes (MFS)[15]. *C. jejuni* has two polar flagella, one at each cell pole, which have an important function not only in motility, but are also responsible for adherence to surfaces, and for the secretion of virulence factor proteins[15,16]. FliD$_{cj}$ is the major antigen in *C. jejuni* and thus a target for vaccine design[17–19].

In this study, we report the structure of the *C. jejuni* flagellar cap complex by cryo-EM. This structure demonstrates that FliD$_{cj}$ is pentameric, with an extensive set of contacts across several residues at the termini, that contribute to stabilizing the oligomeric state. We show that disrupting these interactions has a significant effect on cell motility. We also observe that the full-length FliD protein for both *S. marscecens* (FliD$_{sm}$) and *P. aeruginosa* (FliD$_{pa}$) also form pentamers, with similar dimensions to that of FliD$_{cj}$, indicating that the pentameric state of FliD within the cap complex may be a common motif across bacteria. Finally, we demonstrate that the native *C. jejuni* flagellum filament is 11-stranded, similar to other known flagellum filament structures. These observations allow us to propose a molecular model for the filament-cap interaction, and cap-mediated filament elongation.

## Results

### Cryo-EM structure of the flagellum cap complex.
Existing high-resolution structures of FliD have so far been limited to the head

domain. We therefore sought to characterize the intact FliD protein. To that end, we purified full-length FliD from several species: *C. jejuni* (FliD$_{cj}$), *P. aeruginosa* (FliD$_{pa}$) and *S. marcescens* (FliD$_{sm}$) (Supplementary Fig. 1a). Preliminary negative stain analysis showed that while the complexes formed by FliD$_{pa}$ and FliD$_{sm}$ are heterogeneous (Supplementary Fig. 1b), FliD$_{cj}$ forms a homogeneous complex, suitable for structural characterization.

Next we used cryo-EM to determine the structure of FliD$_{cj}$. The protein forms discrete particles in vitreous ice, and 2D classification confirms that it adopts the dumbbell shaped structure previously reported for FliD$_{se}$ (Supplementary Fig. 2a). In addition, a significant subset of particles adopted top-view orientations, with clear 5-fold symmetry. This allowed us to obtain a structure of the full complex, to 4.7 Å resolution (Supplementary Fig. 2b, e).

The FliD$_{cj}$ complex possesses an overall architecture similar to FliD$_{se}$[5,6], consisting of ten subunits, with two pentamers interacting in a "tail-to-tail" orientation, through the leg domains (Fig. 1a). A pentamer is about 170 Å in height (the decamer is ~300 Å) and 130 Å in width with a 20 Å lumen (Fig. 1c). We note however that the map shows a wide range of local resolution, with the leg domain well defined and with visible density for the large side-chains, while the head domain is more poorly defined (Supplementary Fig. 3a, b). This could suggest that there is some flexibility in the head domains, or that each subunit adopts slightly different conformations, which were averaged out in our map because of the applied 5-fold symmetry. To attempt obtaining a better map for this region of the complex, we performed a focused refinement on the head domain only, leading to a map at 5.0 Å resolution (Supplementary Fig. 2c, e, 3c). Using this focused head domains map, we were able to generate an atomic model for this region of FliD$_{cj}$ (Domains D1, D2, and D3), based on the crystal structure of FliD$_{ec}$ (PDB ID: 5H5V)[8]. We then used the map of the full complex to build the atomic model for the leg domain D0 de novo (Supplementary Figs. 2d, 4, and Table 1).

The FliD$_{cj}$ structure shows that the FliD protomer folds in on itself in a ν-shape, which results in N and C termini next to each other in the leg domain, as illustrated in the topology diagram (Supplementary Fig. 5). The overall architecture, as proposed previously, consists of a D0 domain formed by a long coiled coil, consisting of two helices located at the termini. A four-helix bundle forms the D1 domain. Connected to the D0–D1 leg domains are D2–D3 domains, rich in anti-parallel β-sheets, forming the head (Fig. 1b)[7–9]. This overall architecture is similar to that of the flagellin and hook, and in agreement with the previously reported structures of the FliD head domain[20]. In particular, we note that the D0 and D1 domains in the flagellin are largely structurally similar, and thus synonymous to the D0 and D1 domains in FliD. However, while it was predicted that the D0 domain consists of a two-helix coiled-coil, as present in the flagellin and hook, our structure reveals that the N-terminal 17 residues are extended into a stretch that folds under and behind the monomer, interacting with the preceding subunit via a short β-strand (see below). As a consequence, the C-terminal helix of the coiled-coil is not partnered with the N-terminus, but instead interacts with that of another molecule through hydrophobic interactions, forming the pentamer-to-pentamer interface. This intriguing architecture likely explains the strong tendency of FliD to form tail-to-tail complexes during isolation, as observed in FliD$_{se}$[6] and FliD$_{cj}$ (this study).

We also note that FliD$_{cj}$ possesses a long insert within the D1 helix bundle, not present in other orthologues (Supplementary Fig. 5). Secondary structure prediction indicates that this insert is likely globular, leading to the hypothesis that it forms an additional domain, termed D4. This type of domain insertion is not unusual, and has been observed in other FliD orthologues, as

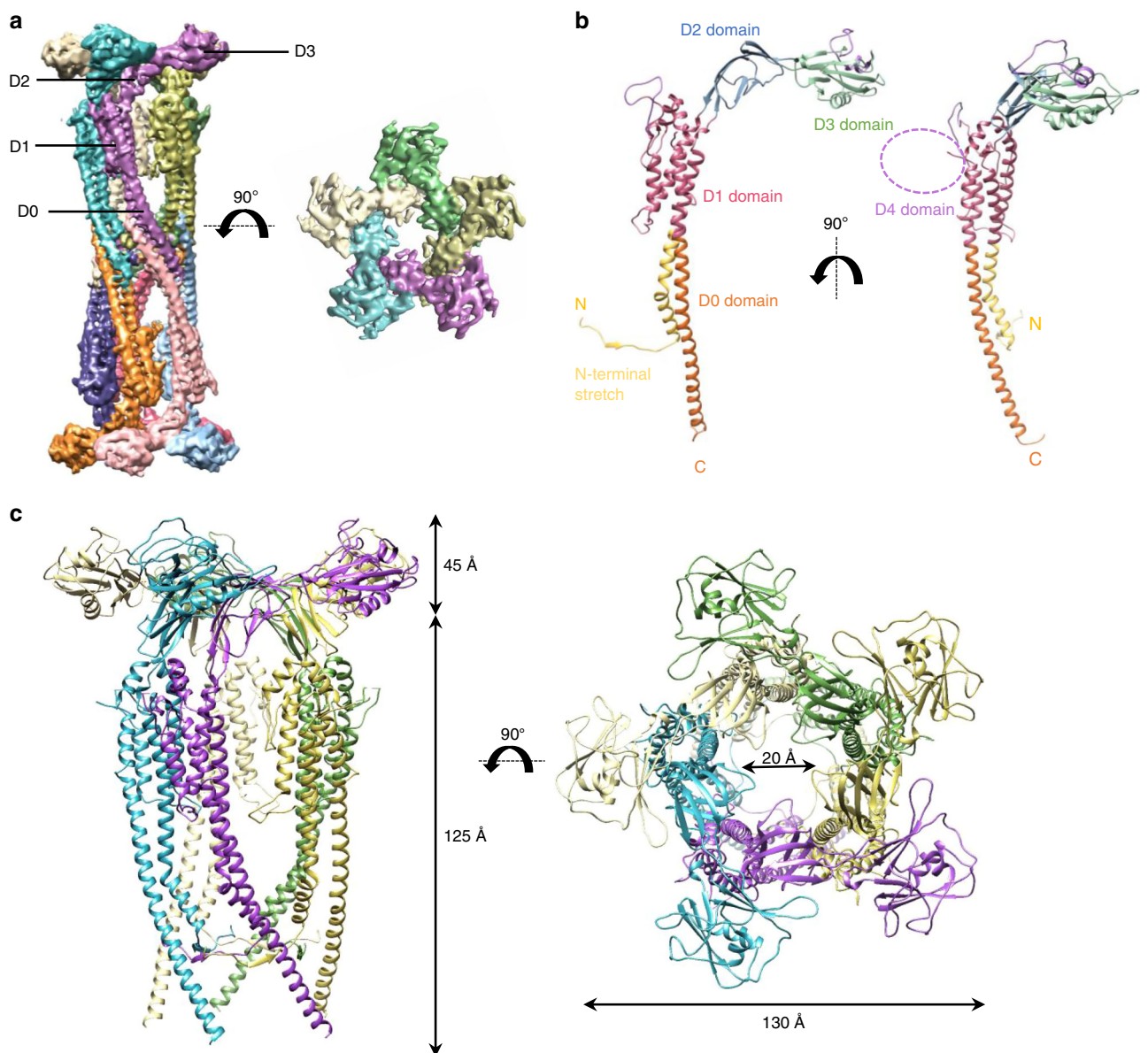

**Fig. 1 Structure of FliD$_{cj}$. a** The cryo-EM map of FliD$_{cj}$, segmented and colored by subunit. A side view of the full complex map is shown on the left, and a top view of the head domain map is shown on the right. Domains are indicated with labeling lines. **b** Cartoon representation of the full-length FliD$_{cj}$ monomer, colored according to domain organization. The purple dotted circle indicates the position of the D4 domain. **c** Cartoon representation of the FliD$_{cj}$ pentamer, corresponding to the intact cap complex, with respective measurements. In our model the D4 domain is not visible as it was not built in. For reference it should be localized as indicated in panel (**b**). Side view (left) and top view (right) are shown and color coded as in (**a**).

well as in flagellin and hook proteins[10,11,21,22]. In our FliD$_{cj}$ map, we were able to observe density for this domain (Supplementary Figs. 3c, 6a), however it is at very low resolution, and did not allow us to build an atomic model. Further 3D classification without imposing symmetry revealed at least four potential distinct positions for this domain (Supplementary Fig. 6b), but the resolution obtained was limited by the number of particles used for reconstruction. The role of this D4 domain is not known, but we postulate that it could be related to FliD$_{cj}$'s capacity to bind to heparin, a feature involved in *C. jejuni* adherence but not observed in other FliD orthologues[4].

**Comparison with other FliD orthologues.** Our structure, at 4.7 Å, represents the highest resolution of FliD full-length complex to date. Nonetheless the crystal structure of the head domain,

corresponding to domains D2–D3, has been reported for a range of species, including *S. enterica*, *P. aeruginosa*, *E. coli*, *S. marcescens*, and *H. pylori*[6–10]. In all orthologues, the structure is very similar, with RMSD values ranging from 1.5 Å to 2.5 Å to that of FliD$_{cj}$ (Fig. 2a). In the *E. coli* orthologue, domain D1 was also present in the structure. It consists of a 4-helix bundle, and this structure is very similar to that of FliD$_{cj}$, with a RMSD of 1.5 Å between the two structures. Nonetheless, we note that the position of D1 relative to that of D2–D3 is different in FliD$_{ec}$ compared to FliD$_{cj}$ (Supplementary Fig. 8a). This could occur because the hinge between D1 and D2 is flexible. Alternatively, it could be due to the fact that in FliD$_{cj}$ the D4 domain protrudes from the D1 helical bundle and thus a more planar conformation, as observed in FliD$_{se}$, may be sterically clashing with D2–D3 in the case of FliD$_{cj}$ (Supplementary Fig. 8b), leading to a conformation distinct to that of other species.

**Table 1 Cryo-EM data collection, refinement and validation statistics.**

| | FliD$_{cj}$ (EMDB-10210) (PDB: 6SIH) | Filament (EMDB-10244) |
|---|---|---|
| Data collection and processing | | |
| Microscope | Titan Krios | Tecnai Arctica |
| Magnification | 36232 | 53000 |
| Voltage (kV) | 300 | 200 |
| Camera | K2 Summit | Falcon III |
| Pixel size (Å) | 1.38 | 2.03 |
| Symmetry Imposed | D5 (full map), C5 (head map) | Helical (Rise: 7.25 Å, Twist: 65.4°) |
| Defocus range (μm) | −1.0 to −2.6 | −0.8 to −2.0 |
| Total dose (eÅ$^{-2}$) | 41 | 45 |
| Number of micrographs | 1223 | 100 |
| Total particles used | 55967 | 71828 |
| Map Resolution (Å) | 4.7 (full map) | 8.6 (symmetrical) |
| | 5.0 (head domain) | 27.2 (asymmetrical) |
| Refinement | | |
| Map sharpening $B$ factor (Å$^2$) | −285 | |
| Model composition | | |
| Non-hydrogen atoms | 37320 | |
| Protein Residues | 4880 | |
| $B$ factors (Å$^{-1}$) | | |
| Protein | 148 | |
| R.m.s. deviations | | |
| Bond lengths (Å) | 0.005 | |
| Bond angles (°) | 0.543 | |
| Validation | | |
| MolProbity score | 1.85 | |
| Clashscore | 8.36 | |
| Poor rotamers (%) | 0.00 | |
| Ramachandran plot | | |
| Favored (%) | 94.13 | |
| Allowed (%) | 5.87 | |
| Disallowed (%) | 0.00 | |

In our cryo-EM map, FliD forms a pentameric architecture, consistent with the low-resolution cryo-EM structure of FliD$_{se}$, with a similar overall architecture consisting of two pentamers in a head-to-tail arrangement. In contrast, crystal structures of the head domains from FliD in several species reported a range of oligomeric states, including tetramer (FliD$_{sm}$), pentamer (FliD$_{se}$) and hexamers (FliD$_{pa}$ and FliD$_{ec}$)[7–9]. When comparing the dimensions of these structures, the diameters of all complexes are similar, around ~140 Å. However, the dimension of the lumen differs significantly between structures, with FliD$_{cj}$ and FliD$_{se}$ having a central lumen of ~20 Å, while FliD$_{pa}$ and FliD$_{ec}$ have a lumen of ~50 Å and ~40 Å, respectively, and FliD$_{sm}$ a ~15 Å lumen (Fig. 2b). Even in the case of FliD$_{se}$, which crystallized as a pentamer, while the overall dimensions are similar to that of the head domains of the FliD$_{cj}$ pentamer, in the S.enterica orthologue the pentamer is flattened compared to that of FliD$_{cj}$ (Supplementary Fig. 8b). This leads us to conclude that there is a difference between our cryo-EM derived structure and crystallography determined FliD structures available to date, not only in the number of subunits but also in the angle of their interaction. It remains to be verified if this corresponds to a general artifact of the crystal structures of D2–D3 domains in isolation or corresponds to a difference in structure between orthologues.

We also note that in some of the previously reported FliD D2–D3 crystal structures, there are some subtle differences between subunits of the oligomer. We attempted to determine if this was also the case in our FliD$_{cj}$ map, by performing further refinement without symmetry. The resulting map did not show any significant difference between the subunits, but the resulting resolution was too low to identify minor conformational changes. Further characterization, using larger datasets and classification, would be required to determine differences in the pentameric subunits of FliD$_{cj}$.

We next investigated the oligomeric state of full-length FliD$_{sm}$ and FliD$_{pa}$, the head domains of which crystallized as tetramers and hexamers, respectively, by negative stain TEM. As mentioned above, these proteins do not form uniform complexes (Supplementary Fig. 1b). Nonetheless, we could observe particles that resembled top views of FliD, which allowed us to perform preliminary 2D classification to determine their lateral symmetry. This revealed that for both orthologues, particles with 5-fold symmetry and similar dimensions to that of FliD$_{cj}$ were present (Fig. 2c). However, in the FliD$_{pa}$ sample we observed additional particles with 6-fold (16.4%) and 4-fold symmetry (58.5%), while in the FliD$_{sm}$ sample there was a large percentage of particles with 4-fold symmetry (52%). The dimensions of the particles in those 2D classes are larger than the FliD pentamer, and therefore we could not conclude if these correspond to alternative oligomeric species, or to other negative stain artifacts and/or non-specific aggregates. However, the presence of pentamers with similar dimensions to that of FliD (25.5% in FliD$_{pa}$ and 48% in FliD$_{sm}$) supports the hypothesis that the native architecture of the cap complex is a FliD pentamer, with contacts at the N-terminus required for FliD to adopt its true oligomeric state.

**Interactions in the D0 domain assist in forming filaments.** As mentioned above, our structural characterization of the cap complex indicates an unusual architecture of the N-terminus, which forms a stretch that wraps around and forms contacts with two adjacent subunits, with mainly hydrophobic residues at the interface (Fig. 3a). In particular, this stretch of residues interacts with the C-terminal α-helix of the adjacent molecule. This is of particular interest since it was shown that the C-terminus contributes to the oligomerization of FliD (as well as interaction with its chaperone)[23]. We also note that both the N- and C- termini of FliD (residues 1–46 and 577–643, respectively, in the FliD$_{cj}$ sequence) are conserved across a wide range of bacterial families within the Proteobacteria phylum as shown in the multiple sequence alignment (Fig. 2d).

To confirm the role of these interface interactions in FliD function, we engineered a C. jejuni fliD knockout strain (ΔfliD), leading to a loss of chemotaxis observed in a soft agar swarm assay. Accordingly, no filament was observed in this strain, and thus this phenotype was attributed to lack of motility (Supplementary Fig. 9a). Genetic complementation by expressing the fliD gene at a distal site on the chromosome fully rescued motility (Fig. 3b, Supplementary Fig. 9a), and we exploited this to engineer point mutations in select highly conserved residues within the interacting interfaces to assess their impact on motility.

Mutation of Leu 9, Phe 11, Trp 614, or Trp 617 resulted in reduction of motility, particularly for mutations to polar residues (F11S, W614S, L9S, or Y617S) (Fig. 3b). This leads us to propose that the hydrophobic nature of this interface may be one of the properties affected by our mutations that have an effect on motility, suggesting that the interaction formed by the N-terminal stretch contributes to FliD function of building the filament and thus cell motility. To verify if motility was affected because the aforementioned mutations prevented filament assembly, we visualized the corresponding bacteria by TEM. All of the mutations still led to bacteria with assembled filaments, of length similar to that of WT bacteria (Supplementary Fig. 9b), demonstrating that the corresponding FliD proteins are still able

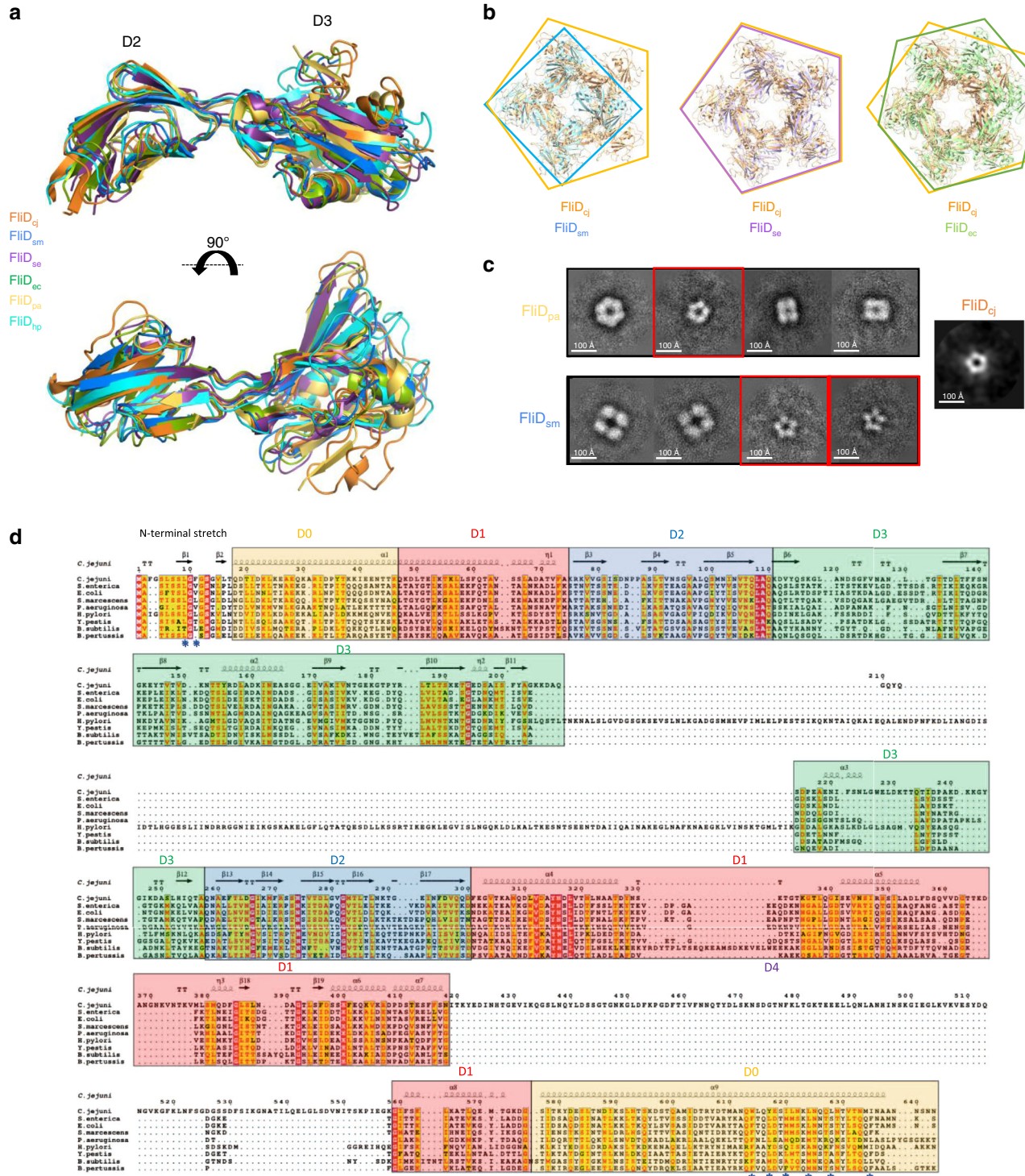

**Fig. 2 Comparison of the FliD structures across bacterial species. a** Overlay of the FliD D2–D3 domains structures from *C. jejuni* (FliD$_{cj}$, 6SIH—this study, orange), *S. marcescens* (FliD$_{sm}$, 5XLJ, blue), *S. enterica* (FliD$_{se}$, 5H5T, purple), *E. coli* (FliD$_{ec}$, 5H5V, green), *P. aeruginosa* (FliD$_{pa}$, 5FHY, yellow), and *H. pylori* (FliD$_{hp}$, 6IWY, cyan). **b** Alignments of X-ray crystallography derived oligomeric structures of the head domains from FliD$_{sm}$, FliD$_{se}$, and FliD$_{ec}$, to that of the FliD$_{cj}$ pentamer. The diameter of the lumen as well as the outer diameter of the capping protein is similar in the pentamer formed by FliD$_{se}$, but significantly smaller in the tetramer formed by FliD$_{sm}$, and larger in the hexamer formed by FliD$_{ec}$. **c** 2D classes obtained from negative-stain data of recombinant FliD$_{pa}$ and FliD$_{sm}$ (Supplementary Fig. 1b). FliD$_{pa}$ shows three different sets of particles: a class with particles showing 6-fold symmetry (~120 Å in diameter), a class showing smaller particles with 5-fold symmetry (~100 Å diameter), and classes of particles with 4-fold symmetry (~80 Å and ~110 Å in classes 3 and 4, respectively). FliD$_{sm}$ shows two different sets of particles: large particles with 4-fold symmetry (~150 Å in diameter), and smaller particles with 5-fold symmetry (~100 Å diameter). The size of the 5-fold symmetry particles in both orthologues is similar to that observed in 2D classification of FliD$_{cj}$. The red squares indicate classes with particles of similar shape and dimensions to the FliD$_{cj}$ structure, with distinctive five-fold symmetry. FliD$_{cj}$ negative stain top view is shown as a reference to the right. **d** Sequence alignment for FliD from different species. The secondary structure elements for FliD$_{cj}$ are shown at the top. The domains are color coded as follows: *Yellow*: D0 terminal domains. *Red*: D1 domain. *Blue*: D2 domain. *Green*: D3 domain. *Uncolored*: D4 domain found only in FliD$_{cj}$. Stars indicate important residues in terminal interactions, for which mutagenesis experiments were performed in Fig. 3.

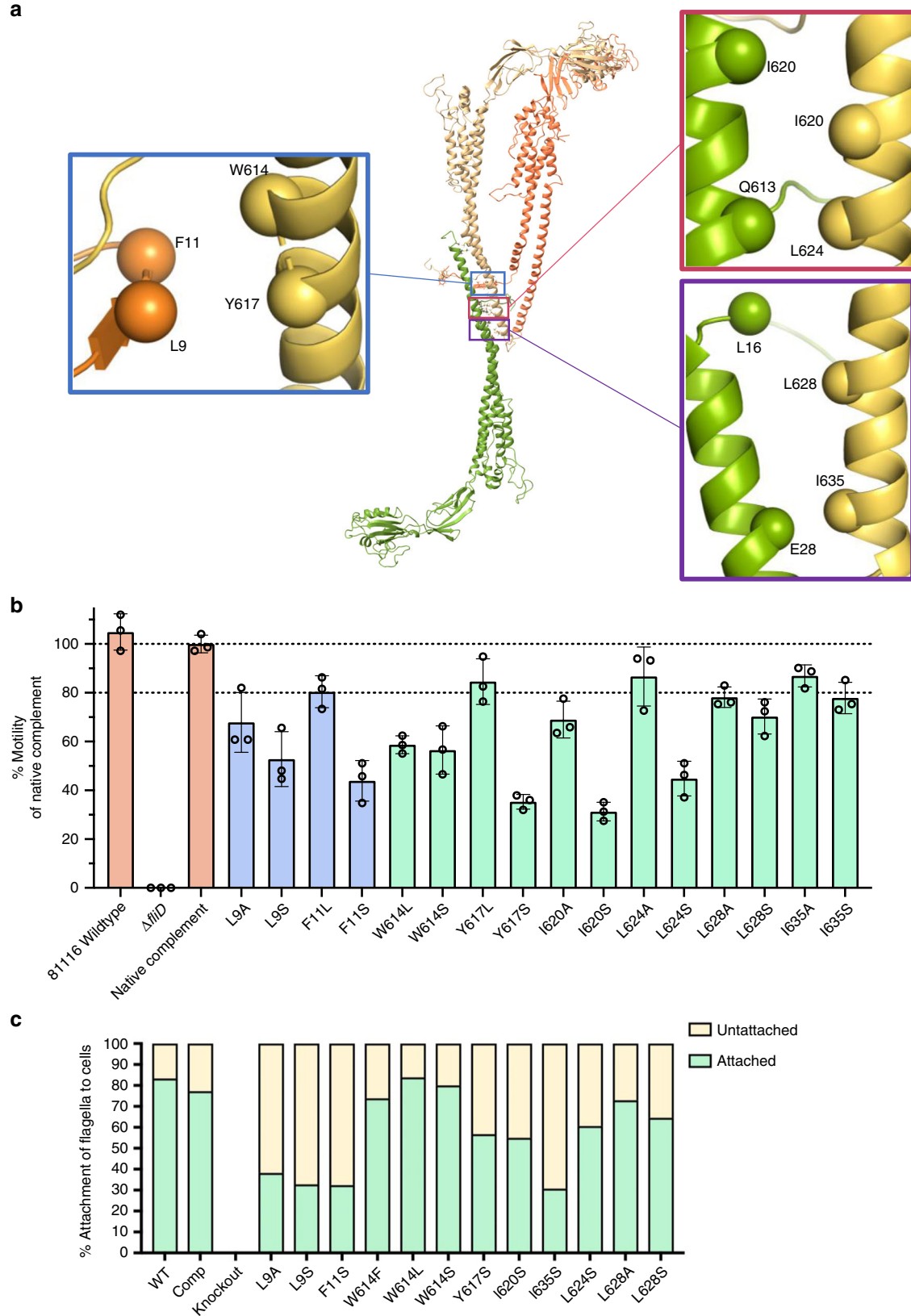

to promote filament elongation. However, we noted that the filaments are less stable in the mutants, with between 60 and 80% of filaments found broken off at varying lengths from the cell body, versus ~20% in WT bacteria (Fig. 3c, Supplementary Fig. 9c). We also note that the N-terminal ~20 residue stretch

corresponds to the secretion signal in flagellar filaments of *S. enterica*, so potentially a similar signal exists for FliD to be secreted through the flagellum T3SS[24]. The observation that in the mutants described above, the filament is still formed, is a confirmation that these mutations did not interfere with FliD

**Fig. 3 FliD$_{cj}$ amino-acid substitutions and their effect on motility. a** Location of the mutated residues on the FliD$_{cj}$ structure, chosen due to their conserved nature and approximate location in the interacting interface. Adjacent subunits (Yellow, Orange) were chosen to represent the interaction of the N-terminus residues with that of the next subunit C-terminus and which might be important in the formation of the pentamer. The residues in the lower C-terminus were chosen to represent the interaction interface with flagellin upon flagellar elongation, as the conserved residues in that region which bind to the bottom subunit C-terminus might mimic the interactions with the flagellin monomer. **b** Motility assay results for point mutants represented as the mean percentage of the native complement strain, based on swarm diameter on soft agar. Controls (WT, deletion mutant and complement) are in orange. C-terminal mutants in light green and N-terminal mutants in purple. Error bars show standard deviation with $N = 3$. The corresponding data points are represented as dot plots. **c** Plot of the percentage flagella attachment to cells calculated from micrographs for each point mutant. Attached are colored green and unattached yellow. Source data are provided as a Source Data file.

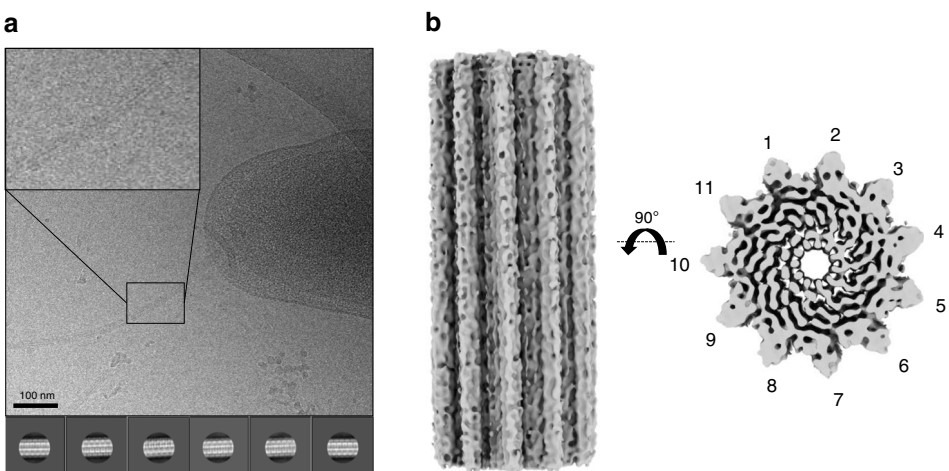

**Fig. 4 Structural characterization of the native *C. jejuni* flagellum. a** Cryo-electron micrograph of the native *C. jejuni* flagellum, used for the 3D reconstruction. 2D classes, generated from ~71828 particles, are shown below. An enlarged image of a bacterial flagellum is portrayed in a panel in the top left corner of the micrograph. **b** EM map of the native flagellum, with helical symmetry applied, to 8.6 Å resolution. A side view is shown to the left, and top view to the right. .

secretion, but rather with its function to promote filament assembly.

The second interacting interface observed in the D0 domain, is formed between the C-terminus of FliD in the pentamer-to-pentamer interface (Fig. 3a). We sought to validate the role of this interface in FliD function. To that end, we engineered a series of mutations in some of the conserved residues forming this interface (Leu 628, Ile 635, Leu 624, and Ile 620) and characterized their impact on motility as described above (Fig. 3b). Mutating these residues impacted motility in a similar manner to that of the residues of the pentameric interface (see above), confirming that the hydrophobicity of this interface also contributes to FliD function in filament assembly.

As our protein complex was recombinantly expressed and assembled in-vitro, we propose that the decameric interface is not physiological. Instead, we propose that the C-terminal region of FliD interacts with the flagellar filament lumen, as supported by previous low-resolution data of intact flagellum tips[5,25–27]. In further support of this hypothesis, co-variance analysis (Supplementary Table 3) reveal that many of the FliD residues (Ile 635, Tyr 617, Leu 624, and Trp 614) have a high probability of interacting with terminal residues of the flagellin.

**A structural model of the *C. jejuni* filament**. Current Cryo-EM structures of various flagellar filaments have demonstrated that they consist of 11 protofilaments, formed by a single protein, the flagellin[11]. The flagellin consists of four domains D0–D3, and can adopt two conformations, termed L and R, leading to two alternative filament structures, left-handed and right-handed, respectively. *C. jejuni* possesses two flagellin paralogues, FlaA and FlaB,

that are ~95% identical to each other, with both required for the formation of fully functional filaments. FlaA and FlaB are highly similar to other flagellins (Supplementary Fig. 10a), except for a ~70 amino-acid insert in D2 that likely consists of a globular insert, as observed in several flagellin orthologues[11]. Surprisingly, a previously published EM structure of the *C. jejuni* filament had reported a seven protofilament arrangement[13]. However, this structure was obtained from a FlaA G508A mutant, in the absence of FlaB, and is at low resolution. It is therefore not clear if this was an artifact and/or wrong interpretation of the data, or if the *C. jejuni* filament indeed possesses a different architecture to other species.

To reconcile this, we sought to determine the structure of the native *C. jejuni* filament, directly from wild-type cells (Fig. 4a, Supplementary Fig. 11). To avoid biases due to symmetry, we initially performed a reconstruction without any helical symmetry using a low pass filtered map of *P. aeruginosa* flagellar filament (EMDB:8855) as a reference. The obtained *C. jejuni* filament map clearly possessed 11-fold symmetry (Supplementary Fig. 10b), despite the low resolution (~ 27 Å, Supplementary Fig. 10c). This demonstrates that the *C. jejuni* flagellar filament consists of 11 protofilaments with a lumen of ~25–30 Å and outer diameter of ~200 Å, similar to that of other bacterial species. We therefore refined the map further by applying helical symmetry, initially with 65.5° twist and 4.7 Å rise as per the values used for the reconstruction of the *P. aeruginosa* flagellum filament[11]. However, this reconstruction did not converge to a map with defined features and a smooth FSC curve. We therefore performed multiple 3D refinements with a search range for both the twist and the rise, which converged on a 65.4° twist and 7.25 Å rise, which allowed us to reach ~8.6 Å resolution (Supplementary Fig. 10c). In

this map the central D0–D1 domains are well resolved, with density for helices clearly visible (Fig. 4b). The density for domains D2 and D3 is visible, but less well resolved. The fact that we can only reach limited resolution is perhaps not surprising, since we likely have a combination of L and R conformations for the flagellin. Other reasons for low resolution may include the amount and quality of the micrographs (collected on a 200 kV microscope), as well as the limited number of extracted particles. Nonetheless, this data conclusively demonstrates that the *C. jejuni* flagellum filament is 11-stranded, and not 7-stranded as reported previously.

Based on this, we used a tomography map of FliD bound to the hook from *B. burgdorferi*[28] (EMDB ID: 0525) to position the previously published structure of the *P. aeruginosa* filament (PDB ID: 5WK6), and the cap complex structure reported here. Three different possible positions of the FliD pentamer in relation to the flagellar filament could be fitted within the density, with one position showing no major steric clashes and with contacts between FliD and the filament consistent with the decameric interface of FliD (Supplementary Fig. 12a, b). This allowed us to suggest a model for FliD-flagellin interaction (Fig. 5, Supplementary Fig. 12a). In this model, the C-terminus of FliD forms broadly non-specific, hydrophobic contacts with exposed regions of the filament, similar to flagellin-flagellin interactions (Supplementary Fig. 12b). A gap between adjacent FliD molecules, on the side of the leg domain, is positioned in a suitable location for the insertion of a flagellin molecule and is the likely site of exit

for nascent molecules. This however remains to be verified experimentally.

## Discussion

In previous studies, evidence suggesting different stoichiometries for the flagellum filament and/or cap complex in different species was based on low-resolution cryo-EM structures, and crystallographic symmetries of truncated proteins. Here we largely resolve this conflicting evidence, by demonstrating that FliD adopts a pentameric stoichiometry in at least three different proteobacteria families, and that the filament of *C. jejuni* is 11-stranded, and not 7-stranded as reported previously. We can therefore conclude that the stoichiometry of these proteins is likely conserved across this proteobacteria phylum, with a 11-to-5 asymmetry between these two different regions of the bacterial flagellum. Based on our structure, we also hypothesize that there is a large degree of plasticity in the interface between the D2–D3 domains of adjacent molecules, and therefore in the absence of D0, a range of interfaces can be trapped in the crystal contacts. We propose that the additional contacts formed by the N-terminal stretch are essential for FliD to adopt its true oligomeric state.

Our structure of the intact cap complex, supported by mutagenesis studies and previous tomography data, suggests that the FliD C-terminal domain interaction with the opposite pentamer in the decameric complex mimics that of the FliD interaction with the filament. Intriguingly, we note that according to this model, the position of the two terminal helices in the D0 domain of FliDcj is distinct from that of the flagellin (and the related T3SS needle)[20,29]: While in the flagellin the C-terminus helix is facing inwards towards the lumen of the filament, in our structure of FliD$_{cj}$ it is facing outwards. This unexpected architecture is consistent with our model of the tip–filament complex, and supported by our mutagenesis analysis. In particular, our data provide evidence to support the hypothesis that exposed hydrophobic residues, both on the D0 domain of flagellin molecules and in the C-terminus of FliD, act as a chaperone-like environment to promote the folding and insertion of new flagellins[30,31]. Nevertheless, further characterization of the cap complex bound to the filament will be required to verify this model.

Based on the results reported in this study, we propose a mechanism for cap-mediated filament elongation, as illustrated in Fig. 5. Prior to assembling the filament, FliD is exported through the lumen of the flagellar hook, and assembles at the end of the hook-filament junction[32]. It is not known whether individual FliD molecules attach (potentially with a helical architecture), or whether it forms a pentamer first, which then docks onto the hook-filament junction. According to our FliD–filament complex model (Supplementary Fig. 12), the FliD cap pentamer fits into the flagellum filament, through interactions between D0 of flagellins and the C-terminus of FliD. We note however that because of the symmetry mismatch, a different set of interactions are formed on each side of the cap complex. It is possible that this is achieved because the interface is largely composed of non-specific hydrophobic interactions. Alternatively, there could be some structural changes of the D0 domain of FliD that our structure does not capture.

New unfolded flagellin molecules are secreted through the filament[31], and our model suggests that they would emerge in a chamber inside the cap complex (1). According to our model, there is a gap between the D0–D1 domains of FliD that is not obstructed by the filament on one side. We propose that unfolded flagellin molecules might exit the complex through this cavity (2)[27]. While there is currently no direct evidence for this, we note that the residues Y315 and N316, which are facing this cavity, are

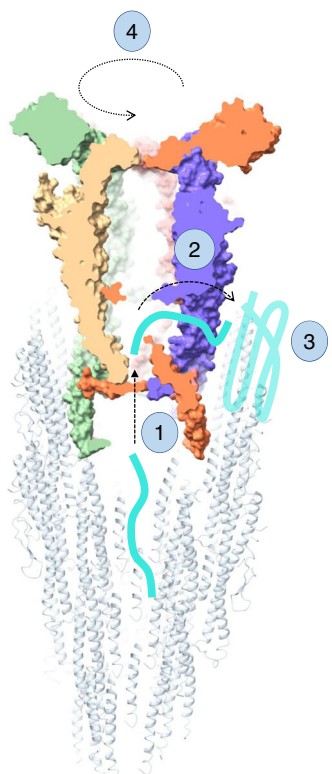

**Fig. 5 Model of cap-mediated filament elongation.** Nascent flagellin molecules are secreted through the filament and enter the cap complex chamber (1). They then exit the cap complex through a side cavity (2), which positions them in close proximity to the site of insertion. There, hydrophobic patches, composed of both exposed flagellin molecules, as well as FliD D0 domain, act as chaperones, and promote flagellin folding (3). Following the insertion of a new flagellin subunit, the cap complex rotates (4), positioning it to have the open cavity towards the next site of insertion.

conserved in proteobacteria (See above), and across all flagellated bacteria. Once outside of FliD, we propose that exposed hydrophobic residues act as a chaperone, and promote flagellin folding in its insertion site (3)[30,31]. In order to accommodate the next flagellin subunit, conformational changes need to occur to open an adjacent binding pocket. We propose that the folding of the new flagellin protomer leads to dislodging of the cap complex, that rotates by ~65° (4), thus positioning an adjacent cavity of the cap complex close to the next flagellin insertion site (Fig. 5)[5,6,8,27]. This hypothesis agrees with previously proposed mechanisms of flagellar elongation[6].

We note that previous studies, based on low-resolution tomography data, have suggested that the D0 domain of FliD might be dynamic, with the leg domains opening and closing to promote filament elongation[5,9,26]. In contrast, in our structure of the cap complex, the D0 domain is rigid, locked in position by the N-terminal stretch. This structure therefore could provide evidence for a different mechanism, which had been proposed previously, whereby the cap complex acts as a rigid cog that rotates during flagellum elongation[8]. Nonetheless, further experiments to characterize the flagellum-cap complex at high resolution will be required to confirm this model, and in particular to observe if the rigidity of the D0 domain, as observed in our structure of the cap complex, is present when FliD is bound to the filament.

In conclusion, we report the cryo-EM structure of the flagellum-cap complex, and demonstrate that FliD across multiple species (FliD$_{sm}$, FliD$_{pa}$, and FliD$_{se}$) forms pentameric complexes. We show that the interface between opposite D0 leg domains in the FliD decamer complex is important for cell motility and formation of a functional filament, and therefore likely plays a role in FliD-filament interactions. We also demonstrate that the *C. jejuni* flagellar filament possesses the same architecture as that of other species. Taken together, these results allow us to propose a model for cap–filament interaction as well as propose a mechanism for cap-mediated filament elongation.

## Methods

**Protein expression and purification**. The genes coding for FliD$_{cj}$, FliD$_{sm}$, and FliD$_{pa}$, codon-optimized for expression in *E. coli*, were synthesized (BioBasic) and sub-cloned into pET28a (Novagen) (Supplementary Table 4). Recombinant proteins were expressed in *E. coli* BL21-CodonPlus(DE3)-RIL cells containing the corresponding plasmids. For FliD$_{cj}$, transformants were grown in LB medium at 37 °C until they reached log phase, and expression was induced by the addition of 1 mM IPTG overnight at 20 °C. For both FliD$_{pa}$ and FliD$_{sm}$, expression was auto-induced in ZYM-5052[33] media at 20 °C overnight. For all three proteins, cells were collected by centrifugation, resuspended in 50 mM HEPES 150 mM NaCl pH 7 and sonicated. The lysate was centrifuged at 14,000$g$ at 4 °C for 45 min. The supernatants were applied onto a 5 ml HisPure™ Ni-NTA resin (ThermoScientific) gravity-based column equilibrated with 50 mM HEPES 150 mM NaCl pH 7 and eluted using a linear 20–500 mM Imidazole gradient. Fractions containing FliD were pooled and applied to a HiLoad Superdex 200 16/600 column (GE Healthcare) equilibrated with 50 mM HEPES 150 mM NaCl pH 7 for FliD$_{cj}$ and 50 mM Tris 150 mM NaCl pH 8 for FliD$_{pa}$ and FliD$_{sm}$.

**Negative-stain grid preparation and data collection**. For negative-stain TEM experiments, ~5 μl of purified protein, or of cell culture in log phase, was applied onto glow-discharged, carbon-coated copper grids (Agar Scientific). After incubating the sample for ~2 min at room temperature, the grids were rapidly washed in three successive drops of deionized water and then exposed to three successive drops of 0.75% uranyl formate solution. Images were recorded on a CM100 TEM (Phillips) equipped with a MSC 794 camera (Gatan) (FliD$_{cj}$ and *C. jejuni* cell cultures) or a Technai T12 Spirit TEM (Thermo Fisher) equipped with an Orius SC-1000 camera (Gatan). Datasets were manually acquired with a pixel size of 2.46 Å pix$^{-1}$, and a defocus range from −0.8 μm to −2.0 μm. The micrographs were processed using cisTEM[34] package, with CTF parameters determined by CTFFIND4[35]. Approximately 3500 particles were picked for FliD$_{sm}$ and 2700 for FliD$_{pa}$ to generate representative two-dimensional (2D) class averages with 330 Å mask diameter.

**Cryo-EM grid preparation and data collection**. For the structural characterization of FliD$_{cj}$, aliquots of (5 μl) of purified protein at a concentration of 1 mg ml$^{-1}$

was deposited onto glow-discharged C-flat holey carbon films 1.2/1.3 200 mesh (EMS). A Vitrobot Mark III (FEI) plunge-freezing device was used for freeze-plunging, using double-blotting[36] with a final blotting time of 6.5 s. Cryo-EM data were collected with a Titan Krios TEM operated at 300 kV and equipped with an energy filter (Gatan GIF Quantum) and recorded on a K2 Summit direct electron detector (Gatan) operated in counting mode. 1223 micrographs were automatically acquired with the EPU software (Thermo Fisher), at a pixel size of 1.38 Å pix$^{-1}$, using a total dose of 41 e$^{-}$ Å$^{-2}$ and with 40 frames per micrograph. The defocus range used for data collection was −1.0 μm to −2.6 μm.

For the structural characterization of the native *C. jejuni* filament, wild-type 81116 strain cell culture grown to OD$_{600}$ = 5 was applied onto glow-discharged C-flat holey carbon films 2/2 200 mesh (EMS). A Leica EM GP (Leica) plunge-freezing device was used for freezing, with a 6 s blotting time. Cryo-EM data were collected on a Technai Arctica TEM (Thermo Fisher) operated at 200 kV and equipped with a Falcon III camera. 100 micrographs were collected using the EPU software (Thermo Fisher) in linear mode, with a pixel size of 2.03 Å pix$^{-1}$, with a total dose of 45 e$^{-}$ Å$^{-2}$ and 1 frame per micrograph. The defocus range used for data collection was approximately −0.8 μm to −2.0 μm.

**Cryo-EM image processing and reconstruction**. For FliD$_{cj}$, processing was done in RELION 3.0[37]. Motion correction was performed with MotionCor2[38], with dose-weighting. CTF parameters were determined by CTFFIND4[35] software. Approximately 2000 particles were manually picked from selected micrographs to generate representative 2D class averages. These classes were used as templates for automated particle picking for the entire dataset. A total of 130000 particles were picked and extracted using a 280×280 pixels box. After multiple rounds of 2D classification, 55967 particles from the best 2D classes were obtained and used to generate an initial model. Following further 3D classification and refinement with D5 symmetry, a final map to 4.7 Å resolution was generated, which was sharpened using PHENIX 1.13[39]. The leg domains were visibly at a higher resolution than the head domains, therefore a mask centering on the head domain was used for further refinement with C5 symmetry, leading to a map of the head domain to 5.0 Å resolution. Further 3D classification of the masked head domain was used to identify 4 different conformations of the D4 domain not resolved in the full map. The processing pipeline is detailed in Supplementary Fig. 7. Local resolution maps were generated using Relion 3.1 Local resolution function (Supplementary Fig. 3).

For the native *C. jejuni* filament, processing was done in RELION 3.0[37]. Motion correction was performed with MotionCor2[38], with dose-weighting. CTF parameters were determined by CTFFIND4[35]. Filaments were manually picked, and particles were extracted using a 4.7 Å rise and 300 pixel box leading to a set of 254041 segments. Multiple rounds of 2D classification gave a final dataset of 71828 good particles which were used for 3D refinement, both with and without imposed helical symmetry. 70 Å low pass filtered map of *P. aeruginosa* flagellar filament (EMDB:8855), in a form of a smooth cylinder with the correct dimensions, was used as a reference for all 3D refinements. Without symmetry, the structure refined to 27.2 Å resolution, but when helical symmetry was applied, the final resolution after further classification and refinement was 8.6 Å, with a 65.4° twist and a 7.25 Å rise. As the FSC curve was not as smooth as expected, a mask was used to exclude the D2 and D3 domain density, and the resulting map of D0 and D1 domains was at a higher resolution with a less structured FSC curve. The processing pipeline is detailed in Supplementary Fig. 11.

**Model building and refinement**. For the D1–D3 domains, a homology model was generated with PHYRE2[40], using the FliD$_{ec}$ crystal structure[8] (PDB:5H5V) as a template. Domains D2 and D3 were fitted into the sharpened head domain map in Chimera[41]. Domain D1 was fitted into the sharpened full FliD$_{cj}$ map in Chimera. Domain D0 was built into the sharpened full FliD$_{cj}$ map using Coot. This model was subjected to iterative rounds of real-space refinement and building in PHENIX 1.16[39] and Coot[42] respectively. The N-terminal stretch was modeled with RosettaES[43], and then the remaining missing loops were modeled using RosettaCM[44] guided by the electron density. The output model was refined once more in Coot to improve the geometry and delete any modeled residues in areas without electron density.

**Cultivation of *C. jejuni***. *C. jejuni* strain 81116 was grown on blood agar plates (Colombia base agar with 5% v/v defibrinated horse blood) in a microaerobic cabinet (Don Whitley, UK) at 42 °C with a controlled atmosphere of 10% v/v O$_2$, 5% v/v CO$_2$ and 85% v/v N$_2$. Where appropriate, the selective antibiotics kanamycin and chloramphenicol were added at 50 μg ml$^{-1}$ and 20 μg ml$^{-1}$, respectively.

**Construction of *fliD* deletion mutant and complement strains**. A *fliD* mutation vector was constructed using NEB HiFi DNA assembly method (E2621, New England Biolabs). Briefly, flanking regions of *fliD* were amplified from *C. jejuni* 81116 genomic DNA using primers fliDmutantF1-R2 (Supplementary Table 1). These flanks were assembled into pGEM3ZF either side of a non-polar kanamycin resistance cassette, amplified from pJMK30 using primers KanF/R (Supplementary Table 1). The final mutation vector was designed such that spontaneous double crossover with the *C. jejuni* 81116 genome would result in the replacement of the

majority of the open reading frame of *fliD* with the kanamycin resistance cassette, allowing a means of selection. For complementation of the mutant, *fliD* was amplified from *C. jejuni* 81116 genomic DNA using primers fliDcompF/R (Supplementary Table 1). The amplified fragment was digested with BsmBI at sites incorporated into the primers and ligated into similarly digested pCmetK plasmid, a complementation vector for *C. jejuni* incorporating flanking regions of the pseudo-gene region corresponding to *cj0046* in *C. jejuni* 11168 to allow insertion into the genome, a constitutive promoter from the *C. jejuni metK* gene to drive expression of *fliD*, and a chloramphenicol resistance cassette. To generate the strains, wild-type *C. jejuni* 81116 was first transformed with the *fliD* mutation vector by electroporation and colonies selected for kanamycin resistance on blood agar plates. The isolated mutant strain was then further transformed with the *fliD* complementation vector and selected for double kanamycin / chloramphenicol resistance.

**Construction of *fliD* point mutants in *C. jejuni*.** Point mutations in *fliD* were constructed by site directed mutagenesis of the complementation vector using the KLD method (M0554, New England Biolabs). Briefly, the *fliD* complementation plasmid was amplified by PCR with divergent primers containing targeted nucleotide substitutions in the forward primer (listed in Supplementary Table 2). An aliquot of the linear PCR product was treated with the KLD enzyme mix to circularize the mutated plasmid while degrading any residual template. The treated plasmids were transformed into *E. coli* DH5α and transformants selected by chloramphenicol resistance. Plasmid was purified from multiple transformants and the *fliD* open reading frame was sequenced to ensure the correct substitution had been introduced without secondary mutations (LightRun sequencing, Eurofins EU). Point mutated complementation vectors were then transformed into the *C. jejuni fliD* mutant strain as above to generate the collection of point mutant strains.

**Motility assays.** Overnight growth of *C. jejuni* on blood agar plates was harvested and resuspended in phosphate buffered saline to an optical density at 600 nm of 1.0. 0.5 μl aliquots were then injected into semi-solid agar plates (0.4% w/v agar, 3.7% w/v brain heart infusion) containing $5 \times 10^{-3}$% triphenyl tetrazolium chloride, a redox dye which allows clear visual assessment of growth. The diameter of growth was measured after 16 h of incubation. The data presented in Fig. 3b was collected in triplicate for each sample in three different experimental sessions.

**Flagella attachment visualization via negative stain TEM.** The WT, knockout, complement and all point mutant strains were taken from the same sample used to inject the agar plates and the point mutant flagella attachment was determined through imaging grids at ×700 magnification at about 20 micrographs per mutant containing cell count from 30 to 100 cells. The filament was counted as a "fla-gellum" when it had 2 or more inflection points (whether it was attached or on its own in solution). Filaments with less inflection points were considered fragments and not included in the calculations. The percentage of attachment was calculated as a proportion of the total flagella observed per mutant. There was no distinction between fully unattached flagella and partially broken off filaments.

**Generation of FliD–filament complex model.** The low-resolution tomography map of FliD bound to a hook in FlaB knockout mutant of *Borrelia burgdorferi* was used as a volume for docking atomic models (EMDB: 0525)[28]. The pentamer FliD$_{cj}$ model (this study) was used for the region corresponding to the cap, and the *P. aeruginosa* flagellum model (PDB: 5WK6) was used for the filament, as our filament map was not high enough resolution to obtain an atomic model. The FliD$_{cj}$ pentamer could be fitted in three positions, as shown in Supplementary Fig. 12. The central position is clearly the model with least steric clashes between FliD and flagellin, and we therefore used this for the mechanistic model shown in Fig. 5. We note that there is some extra density of the tomography map above the density attributed to D2–D3, which could correspond to the D3 head domain present in *B. burgdorferi* FliD.

**Sequence analysis.** FliD$_{cj}$ sequence in Fig. 2d was aligned to FliD sequences from other organisms using T-Coffee server and visualized with ESPript 3.0[45]. The same was done for the flagellin (FlaA$_{cj}$) sequence in Supplementary Fig. 10a. The co-evolution analysis between FliD$_{cj}$ and flagellin (FlaA$_{cj}$) was performed using RaptorX Complex Contact prediction server[46].

**Reporting summary.** Further information on research design is available in the Nature Research Reporting Summary linked to this article.

## Data availability
The map for FliD$_{cj}$ is available at EMDB with accession code EMD-10210, and the atomic model is available in Protein Data Bank with accession code 6SIH. The map for the native filament is available at EMDB with accession code EMD-10244. The source data underlying Fig. 3b, c and Supplementary Fig. 1a are provided as a Source Data file. Relevant data supporting the findings of this study, plasmids and bacterial strains are available from the corresponding authors upon request. Source data are provided with this paper.

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

## Acknowledgements

This work was funded by a UK Biotechnology and Biological Sciences Research Council (BBSRC) grant (BB/R009759/1) to J.R.C.B. N.S.A. was recipient of PhD scholarship from The Global Strategic Alliance at the University of Sheffield. A.J.T. was funded by a BBSRC grant (BB/R003491/1) to D.J.K. We thank the members of EM facility for their essential assistance and microscope access, and we acknowledge the members of Prof. Per Bullough's laboratory for fruitful discussions. Cryo-EM data for FliD$_{cj}$ was collected at the UK national Electron Bio-Imaging centre (eBIC), proposal EM19709-1. The *C. jejuni* filament data was collected at the University of Sheffield Electron Microscopy Facility.

## Author contributions

N.S.A. and J.R.C.B. conceived the project and designed the structural experiments. A.J.T. and D.J.K. designed the *C. jejuni* cloning, mutagenesis and motility assays. N.S.A. performed the protein purification, Cryo-EM data collection and processing, as well as the negative stain experiments. A.J.T. performed the *C. jejuni* mutagenesis and motility assays, together with N.S.A. S.T. helped with data collection and setup of electron microscopy facility. D.F. and F.D. refined the FliD$_{cj}$ atomic model with Rosetta. N.S.A. and J.R.C.B. wrote the manuscript, with contribution from all the authors.

## Competing interests

The authors declare no competing interests.
