## [Peer Review File · Nature Communications]

Reviewers' comments:

Reviewer #1 (Remarks to the Author):

Dear Editor,

As this paper was available on BioRxiv we took the opportunity for discussion and review of the manuscript within the group. The following people should therefore also be recognised as having contributed to this review:

Dr. Katharina Braunger
Dr. Steven Johnson
Dr. Emily Furlong
Mr. Magnus Jeffrey
Ms. Natalya Davies

Strengths:

The manuscript describes the first higher resolution structure of a full-length flagellar cap complex. As such it presents a major step forwards for the field and will add significantly to our understanding of flagellar assembly.

The manuscript also presents strong evidence that the *C. jejuni* flagellar filament is 11-fold in contrast to earlier work, supporting the idea of conserved architecture across all flagellar filaments. Finally the authors present a characterisation of novel point mutations, based on their structure, which impact bacterial motility.

These are all important results and mean that the work is definitely worthy of publication in Nature Communications – however there are some issues with presentation of the data and details of interpretation that need to be addressed. Most of these issues are to do with interpretation of relatively low resolution volumes and it is a shame, that with the size of the datasets used, it is impossible to understand whether the current resolution is limited by inherent flexibility of the objects or simply reflects lack of data.

To be addressed:

The resolution obtained for the cap complex is modest (and stated with rather unjustified precision). We note that the cap complex maps were calculated in Relion 2.0, while the filament work was carried out in Relion 3.0. Have the authors attempted the cap reconstruction in Relion 3.0 in order to take advantage of the more robust Bayesian particle polishing and per-particle CTF refinement? This may give the boost needed to answer resolution limited problems. In addition, all refinements imposed 5-fold symmetry which could blur out features in the head region. Previous crystallographic structures of pentameric cap heads display significant deviations from true 5-fold symmetry (FlIDSe, 5H5T and FlGDSt, 6IEE). While particle numbers may prevent robust reconstruction with C1 symmetry, the issue should be discussed in the manuscript.

Given the current modest resolution, key features that are later used to construct the functional model need rather better justification in terms of showing the experimental volume. Specifically Local Resolution maps should be shown so that the resolution of the map in the critical termini can be more easily appreciated.

From the figures shown (both main text and supplement) it is clear that there is sufficient resolution to accurately place complete models, but none of the volumes shown give clear support for resolution of side chain orientations. In fact, many of the figures shown in the supplement show side-chain like densities that are unoccupied and bulky side chains lying out of density. If the authors wish to use specific side chain orientations/interactions in generating their models they should support the location of those side chains by demonstrating that the density resolves their location. Even in the helical portions of the N- and C- termini it is unclear where there is sufficient resolution to accurately determine the helical register and the methods do not describe how the long helices were built. In light of this panels with the detailed side chain interactions shown in Figure 3a need much stronger support from the experimental data or co-evolution analysis. The placing of the N- and C-terminal helices with respect to the helical axis of the filament (N- in) is interesting, as all high resolution flagellar and needle structures have so far shown conservation of assembly with the C-terminal helix forming the inner surface of the filament. While cap structures may be assumed to assemble via a different mechanism, especially as the hook cap

FlgD only has an N-terminal helix, the observation of the C-terminal helix on the outside is still noteworthy. However, particularly in light of the fact that early, low resolution models of the type three needle (our own included) interpreted low resolution density as N-terminus in and were only corrected a much higher resolution, the volume justifying this placing of the helices should be shown and this feature of the structure discussed.

The section comparing homologues has a number of problems.

The negative stain EM data would benefit from a reference structure to orient the reader, either in the form of equivalent 2D from the FlIDCj structure or a scaled model. This would hopefully make it clearer that the much larger 6-fold class in the FlIDPa and the large 4-fold classes in the FlIDSm are likely contaminants.

The text comparing crystal structures and Fig S4b seems to contain wrong species descriptions.

The text on line 174 refers to an E. coli orthologue pentamer, but this should be Salmonella.

Similarly, the legend to Fig S4b refers to FlIDSm, when the overlay is with FlIDSe (5H5T) and refers to D1 domain overlays when only D2-D3 domains are shown. Clarification should also be provided for which subunit of the FlIDSe pentamer was used for the overlay in Fig S4b. 5H5T shows significant deviations in angles between the 5 copies are therefore the choice of chain would impact the angles seen in the overlay of the whole objects.

The resolution of the filament structure is surprisingly poor for a filamentous structure with this number of boxed regions. Have the authors performed any 3D classification or other analysis that might help explain this? Why is the FSC curve so structured?

Motility assay and point mutations.

The description in the main text suggests that the effect of the point mutations is to make the filaments brittle. These referees can see no mechanism by which mutations in the cap can lead to assembly of filaments that more readily snap at the cell body, i.e. the opposite end to the cap structure, and it is perhaps surprising that mutations within the cap lead to either no filament or an entirely normal length filament on the bacterium. What molecular events lead to these effects? Are the filaments in the supernatant truly broken off or are have they been nucleated within the supernatant by cap structures that have fallen off the filament? Figure S5c should be brought into the main manuscript.

Since none of the point mutants are as fatal in effect as the knockout, have the authors made any double mutants that are?

The statement that the mutations targeting the proposed filament assembly site are less impactful is not supported by Fig 3b. I620 and L624 are just as important as any of the proposed N-terminus interacting residues. Care needs to be taken in carving up these proposed binding sites. Y617 is only 1 turn of helix up from L620 and has a very similar mutagenesis profile, so categorically attributing the affect of the mutations to one or other interaction is difficult, especially as the proposed flagellar filament interactions can't be a particularly good mimic of the native binding site (see next point).

With this in mind, inspection of figure 4d shows that in the constructed model of the cap on the filament, all of the cap helices lie within the filament ones. However inspection of Fig 1a shows that the interactions that build the decamer are side-ways interactions between helices at the same radius and it is therefore unclear how the interactions that build the decamer are a proxy for the any of the interactions that lead to assembly on the filament. The fact that all the interactions will be different is also not acknowledged in discussion in the text.

Proposed mechanism – there were various aspects of the proposed mechanism that these referees feel either need further explanation, further experimental support or perhaps simply less absolute language in their description.

Is the closed pentameric complex the physiologically relevant state? Whilst the idea that the object seen in isolation is the one that will be found at the tip of the growing filament is attractive, the authors should be careful in how firmly this is stated. The mutations do not prove the physiological relevance of the complex, only that the termini are important for function. These referees feel that unless the cap can be visualised in situ, or the hypothesis supported in another way such as by construction of fully functional disulphide locked cap complexes, then the language used should be moderated.

The authors do not really discuss how, during assembly on the filament, the cap complex would form this planar complex rather than a helical one. The cap must initially assemble at the end of the hook filament junction which shows helical symmetry, and the initial binding event must be tight enough to allow capture of the cap monomers being secreted. The cap may initially therefore follow the helical symmetry and then rearrange into the planar structure observed here. One assumption would presumably be that this object is much more stable than 5-copies in a helical

arrangement? For all other stable flagellar/type 3 complexes co-variation of residue pairs has been able to provide strong support for threading of sequence in modest resolution volumes. Can the authors provide such data to support construction of their model?

How was a completely symmetric, planar pentamer docked onto the helical filament, i.e. how were decisions about rotational position and height within the assembly made? Surely one problem with this symmetry mismatch is that every FliC/FliD interaction would be unique. The authors then state that "Because of the symmetry mismatch, this interaction is not present on one side of the cap complex". Presumably this is reflected by the arrow in Fig 4d (although this is not referenced in the figure legend), but there isn't a clear cavity in the figures shown and surely any such cavity would always be smaller in this model than in a model that allowed flexibility of the legs.

The final model invokes rotation of a rigid cap pentamer but various elements of the proposed mechanism need further explanation, including details such as why folding of flagellin leads to a 35° rotation of the cap ($360/11 \neq 35$ nor is $360/5$)? The major difference between this model and earlier ones (which also invoked rotation) is the idea that the cap legs are locked by interactions within the cap complex and that their flexibility is therefore not a key element of filament assembly. As noted above, further evidence either in the form of strong co-variation of the residues involved in assembling the cap complex or by demonstrating that covalently cross-linking the complex does not perturb function would be required to firmly refute any role for flexibility.

Yours sincerely,

Prof. Susan Lea, FMedSci
Sir William Dunn School of Pathology
University of Oxford

Reviewer #2 (Remarks to the Author):

The authors present the first sub-nanometer structure of the in-vitro purified cap complex from *Campylobacter jejuni*. The main finding is that it has 5-fold symmetry together with some evidence that the identified interactions are important for motility. Furthermore, they have reconstructed the native filament and show that it consists of 11 protofilaments, which corrects previous claims by another group. Despite the limited resolution, they were able to build an atomic model and propose a hypothesis for filament-cap interaction and elongation.

The manuscript is well written and generally laid out clearly. However, the author's conclusions are often somewhat far-reaching, which may not be justified by the presented evidence. I will provide detailed examples below. The discussion is unusually short and not well separated from the narrative in the results section.

The presented results are overall novel and of good quality. The finding that the filament is 11-stranded is a major result on its own and became an integral part of their interaction model, yet it is not described in great detail. The elongation model is very speculative and rests mainly on the author's imagination. The results section should be simplified to contain the findings without adornment – insights and conclusions should be moved to the discussion, which should be expanded. The reality is that the resolution of the structures presented in this study is at the limit for making functional claims at the molecular level. Nevertheless, the main findings represent an advancement and can stand on their own.

Detailed comments.

42 apparatuses

Like 'status', the Latin plural of the noun 'apparatus' is 'apparatus' – I would refrain from anglicizing it and replace it by "molecular machinery" or "mechanisms", etc.

79 "we show that these interactions are essential for cell motility"

The presented evidence (Fig 3b) indicates that mutation of these residues affects motility.

However, they are not essential, since these mutant strains are still motile.

82 "the pentameric state of FliD within the cap complex is likely universal"

The term "universal" is not justified based on a sample size of 3 species. Should provide broader evidence if this statement is kept – e.g. sequence alignments of the major bacteria within the kingdom. Or tone down the phrase to something like "...may be a common motif within the kingdom of bacteria"

105 While the FSC threshold can be cited to arbitrary precision, $1/100\text{\AA}$ is nonsense. Change to 4.7\AA .

113 "suggests that complex is dynamic, which a hinge between leg and head domains"

The lower resolution in the head domain could be a result of alignment errors or the averaging of multiple still unseparated classes and does not necessarily suggest a dynamic structure. What is the character of this hinge? Where is it located? How did the authors arrive at this conclusion?

118 "build the atomic model for the leg domain de novo"

No validation of the new model besides Ramachandran statistics is provided. I suggest calculating a Fourier shell correlation between map and the synthetic model-derived density and add this cross-correlation curve to Fig. S6b. That curve would reveal up to which resolution the model detail is consistent with the experimental map. Any interpretation of molecular details beyond this resolution is not justified.

130 "N-terminal residues are extended into a stretch that folds..."

at 4.7\AA resolution, the tracing of a single polypeptide chain is expected to be very difficult. Please provide examples of this map feature together with the fitted model. What were the B-factors of the refined amino acid residues in this region?

144 "low res = flexible"

Again, this is not a universal truth. From the sharpened maps in Fig.S3b it appears that the map features in that region are below nanometer resolution. How many particles were included in the sub-classification shown in Fig.S3c? At such low resolution and presumably after low-pass filtering, the iso-contour of the envelope of domain D4 is not indicative of distinct conformations. I suggest omitting this conclusion about flexibility altogether and leave it at the fact that additional density corresponding to D4 was observed.

152 "first high resolution structure" of an "intact FliD protein"

Since the "resolution revolution", 4.7\AA is not "high res" anymore, sorry. If superlatives are needed, replace with "our structure, at 4.7\AA , represents the highest resolution of this complex to date", or similar. The 4.7\AA structure is furthermore the E.Coli-expressed, in-vitro purified cap dimer. I doubt this qualifies as "intact", whatever this means. Replace e.g. with "full length", if the case.

156 RSMD

161 "dramatically different ... D1 position ... suggests that the hinge between D1 and D2 is flexible" These are two different classes within the phylum of proteobacteria (gamma and epsilon)! This conclusion and the site of the hinge region are speculative at best.

175 "large degree of plasticity between D2-D3"

Again, while this observation is interesting, it is not a direct observation within the FliD(cj) sample, to which the statement applies. The only conclusion that can be drawn from this comparison is that the two structures are different. Such speculations should go to the discussion section, not in results.

Fig.2c no scale bar

Important, since the legend talks about size comparison. Please provide absolute dimensions.

207 "we also note that N- and C- termini of FliD are highly conserved across species..."

Which species? The examples given are from different classes. Across phyla? Within the domain? Provide sequence alignment or reference to support this statement.

Fig.3a need to show a superposition of the electron potential map to support the fitting of these side chain rotamers.

Fig.3b What was the number of times the motility tests were repeated for each of these mutations? There are error bars for the standard deviations, but N is not provided in the legend nor in the methods section.

219 "this confirms that the hydrophobic properties are critical for motility"

While this is an interesting experiment, this conclusion is not valid as stated. Hydrophobic residues were mutated to A,L,S and 20-50% reduction in motility was detected. However, whether this effect is due to the hydrophobic character of the side chains or other reasons (steric effects, a variation in filament length due to altered assembly kinetics, altered filament geometry, etc) is unclear. Furthermore, FliD is not the chief determinant for motility, but it regulates filament assembly, as stated by the authors. The effect should simply be described here. Any interpretations should be toned down and moved to the discussion section.

236 "evidence from tomography indicates that this interaction is not physiological. However, since it is observed both in FliD(cj) and FliD(se) we postulated that it mimics the interaction between FliD and the filament"

Both FliD(cj) and FliD(se) were in-vitro assembled cap dimers, which are not physiological. End of story. Any conclusions in this direction are pure imagination.

266 "to avoid biases due to symmetry, we initially performed reconstruction without any helical symmetry applied"

How was the initial reconstruction created? This is not described in the methods section. It is remarkable, that for a straight helical filament such evenly distributed angular sampling could be achieved. The cross section in Fig.6 shows very little anisotropy. What structure was used as initial alignment reference? More details are required.

270 "we refined the map further by applying helical symmetry, with 65.4 deg twist and 7.25A rise"
How were these initial helical parameters derived? Are these values the refined parameters after helical refinement with rellion? If so, what were the starting values?

274 "the fact that we can reach only limited res is not surprising, since we have L and R..."
Or that there were not enough particle images, or the image quality wasn't good enough, or alignment and classification were incomplete,..." There are many reasons for limited resolution. Mix of conformations is only one of many possibilities.

285 and Fig. 4. Why use the P.aeruginosa filament and why not show the 8.5A C.j. map together with FliD(cj)? Any "atomic" model over-interprets the present data. The new structures in this paper are nice, although at limited resolution. Simply show them as is.

It would further be prudent here to expand upon the 11-stranded filament structure and describe it in greater detail.

295 "in a range of species"

in 3 species, as described in this paper

303 "hydrophobic residues act as chaperonin-like environment to promote folding"

This idea has been put forward before, as referenced, and should not be presented as a novel hypothesis.

306 "a universal mechanism for cap-mediated filament elongation"

The claim "universal" is based on 3 examples of structures at intermediate resolution, without any additional evidence, not even sequence alignments of the supposedly important hydrophobic residues.

310-315 This is pure imagination of the authors. Please provide evidence for each of these steps or rephrase this discussion paragraph as a literature review with cited references.

316-325 rotation model

"our structure does not support this model, as we show that the N-terminal stretch of FliD is essential for filament elongation and maintains the leg domains in a rigid position"

Where do the authors show that the N-terminal stretch is essential for filament elongation?

The term "rigid" is derived from the fact that the leg domains have locally higher resolution in the

FliD dimer. However, this may be a direct result of the dimer interaction and may not be physiological. The same applies to the conformation and interactions of the N-terminal stretch. None of these observations support the conclusion that the function of the N-terminal stretch is to keep the legs in a rigid position – it could as well be interacting with flagellin in the physiological state. Therefore, the conclusion that these data are incompatible with a dynamic leg model is far-fetched.

Reviewer #3 (Remarks to the Author):

This straightforward and insightful article describes the author's work to better understand the mechanism of bacterial flagella filament elongation. To do this, they focus on the role of FliD, the "cap" protein believed to ride the tip of the elongating flagellar filament. Using the well-established alternatively flagellar model organism, *Campylobacter jejuni*, the authors purify a protein complex and determine its structure by cryoEM. Furthermore they use their structure to propose mechanistic hypotheses about filament that they subsequently test by mutagenesis and monitoring of swarm plate motility. The author's structure suggest a mechanism for FliD oligomerization and how it chaperones flagellins to polymerize into the flagellar filament.

The field has a number of controversies, centring around the belief that different organisms build flagella with different oligomeric states in different components (for example, it has been suggested the *Campylobacter* filaments are formed of 7 protofilaments in place of the normal 11, and that FliD oligomers are usually pentameric but sometimes tetrameric or hexameric). This study convincingly puts to rest this unlikely suggestion by showing that *Campylobacter* has a flagellar filament with 11 protofilaments and a pentameric FliD cap.

The writing is overall good and clear, although at times word choice is somewhat confusing, and the figures I found to be less illustrative of the important findings than I had hoped. But overall an interesting and important study.

There is one major point that will need addressing before acceptance for publication, however, beginning around line ~226: The authors show that a series of point mutations in the N-terminal portion of FliD still allows for the synthesis of a full-length, WT filament, but that these filaments are more "brittle" than a filament that polymerizes under a WT FliD cap. The evidence for this claim is the increased number of broken filament pieces present in negative-stain micrographs of these FliD mutants relative to WT.

The presence of these filament pieces (along with the apparent WT lengths of the FliD-mutant filaments) must mean the defect caused by the FliD mutations is affecting the filament itself (i.e. if the defect were at the FliD/filament interface, as would be expected, one would expect shorter filaments but not a greater number of broken filaments than WT). Thus, the authors seem to be implying that the pentameric FliD cap at the very end of the multi-micron filament is modulating the flexibility/rigidity of the filament (and possibly the hook?) as a whole. If true, this is a novel and, to my knowledge, completely unprecedented role for FliD. If it is true that the filaments are breaking at the rod-hook interface, and not simply at random along the filament, this finding is even more significant (and confusing).

However, the sole piece of evidence for this claim is the observation of more filament pieces on grids by negative stain EM. But, even with WT *C. jejuni* and with very gentle pipetting using cut tips to avoid filament shearing and frothing the cell suspension, there will still be a significant number of filament pieces on the grid, and there is a lot variability day to day, grid to grid and grid-square to grid-square in the number of disattached and broken filaments seen by negative stain TEM (at least in our hands).

The authors will need to provide additional experimental evidence for the increase in brittleness in the FliD N-terminal mutants. This could possibly be done by, e.g., western blotting or Coomassie staining for flagellin in the culture supernatants of the various mutants, among other methods. It would be welcome if the authors could also speculate a mechanism for how the FliD cap is tuning the rigidity of the entire flagellar filament.

Additionally, the authors state that 60-80% of filaments in the mutant backgrounds are disattached vs. 20% for WT. This needs clarification. What exactly are the authors quantifying and how are they doing it? Does this mean that 60-80% of cells have an entire filament (with hook?) laying directly next to the pole of a non-flagellated cell, and that this filament unambiguously snapped off that specific cell? If so, how do the authors know it came from that cell? Are the filaments disattaching at the hook-rod interface? How do the authors know that an aflagellate cell had a filament break off and simply hasn't constructed a filament in the first place? How do they know whether a cell with a stubby filament has a broken filament vs. being in the middle of constructing a new, unbroken filament?

Finally, from the micrographs provided, it appears to me that some of the FliD mutants (e.g. L628A, I620S, L9S) ARE making shorter filaments than WT. Have the authors attempted to measure filament lengths of various mutants? Even if the difference in length were 10-20%, this would be significant.

I also have a number of minor suggestions:

- * Line 22: use of the word "allow" is a strange choice, and may be directly misleading to newcomers to the field. "Allow" suggests that bacterial flagella relieve some unstated inhibition of motility. Suggest a word like "drive" or "facilitate".
- * Line 31: the 11-protofilament flagellar filament is important, and my feeling is that the authors could claim more credit for bringing this to light.
- * Line 69: "often the most common case" is strange wording. "the most common case" is singular, so how can this be "often"? Suggest removing "often".
- * Line 71: suggests two polar flagella, two at each pole. Suggest clarifying with "one each at each pole" or similar.
- * Figure 1: I was somewhat disappointed not to be helped to understand this figure better, which looks very nice, but could communicate key findings more clearly. I'd suggest a topology map of the protein fold would be a helpful addition. Panel A would benefit from labeling (with labeling lines) the four domains in one of the monomers). Panel B would benefit from colour-coding by sequence position from N- to C-terminus instead of by domain: the domain labelling already present is more than sufficient, and one of the interesting features of the fold is that it folds back upon itself, with D0, D1, etc composed of sequence-separate segments. Finally I would like to see the position of D4 labelled explicitly in the models, especially panel C (as in, e.g., Fig S3). At the moment the molecular model suggests, misleadingly, that D4 does not exist.
- * Line 115: I'd like to see more description of how the authors combined these two partial maps so that they could build an atomic model.
- * Paragraph of line 121: I think the readers deserve some help here in outlining that "D0" and "D1" are structurally similar between flagellin and FliD, and that this likely reflects that they are homologous, and that "D0" in flagellin is synonymous with "D0" in FliD. Don't make them have to figure it out for themselves.
- * Line 131: "interacting with the preceding subunit" really deserves a dedicated panel in Figure 1 to illustrate this.
- * Line 156: "RMSD"
- * Line 181: "verify": if the authors set out to "verify" their hypothesis, this suggests to me that they were "trying" to confirm their hypothesis. A more satisfying word would be "test".
- * Line 188: what percentage? Needs to be directly stated in Results.
- * Line 189: what percentage? Needs to be directly stated in Results.
- * Line 190: "significantly": by what statistical test? By how much? I suspect the authors mean to say "substantially".
- * Line 217: "reduced motility": I had to search in the figure legend to find, specifically, what the authors mean by "motility". They mean swarm diameter on swarm agar, which does not directly reflect motility—it might reflect chemotaxis. Suggest using more specific language so as not to mislead the reader. (though I have no doubt that, in this case, it reflects motility).
- * Line 253: "homologues" is more accurately "paralogues" (I'd congratulate the authors on correct use of the term "ortholog(ue)" throughout!)
- * Paragraph of line 264: did the authors try imaging a straight mutant, or the G508A mutant? I'm not asking that they do this if they haven't already, but if they did, I'd be very curious.
- * Figure 4a: could have close-ups of filament
- * Fig 4D, 5: the mixture of density map and cartoon are confusing and don't help the reader

understand the model. Suggest standardizing on one or the other?

Reviewer #1

Dear Editor,

As this paper was available on BioRxiv we took the opportunity for discussion and review of the manuscript within the group. The following people should therefore also be recognised as having contributed to this review:

Dr. Katharina Braunger
Dr. Steven Johnson
Dr. Emily Furlong
Mr. Magnus Jeffrey
Ms. Natalya Davies

Strengths:

The manuscript describes the first higher resolution structure of a full-length flagellar cap complex. As such it presents a major step forwards for the field and will add significantly to our understanding of flagellar assembly.

The manuscript also presents strong evidence that the *C. jejuni* flagellar filament is 11-fold in contrast to earlier work, supporting the idea of conserved architecture across all flagellar filaments

Finally the authors present a characterisation of novel point mutations, based on their structure, which impact bacterial motility.

These are all important results and mean that the work is definitely worthy of publication in Nature Communications – however there are some issues with presentation of the data and details of interpretation that need to be addressed. Most of these issues are to do with interpretation of relatively low resolution volumes and it is a shame, that with the size of the datasets used, it is impossible to understand whether the current resolution is limited by inherent flexibility of the objects or simply reflects lack of data.

To be addressed:

The resolution obtained for the cap complex is modest (and stated with rather unjustified precision). We note that the cap complex maps were calculated in Relion 2.0, while the filament work was carried out in Relion 3.0. Have the authors attempted the cap reconstruction in Relion 3.0 in order to take advantage of the more robust Bayesian particle polishing and *per-particle CTF refinement*? This may give the boost needed to answer resolution limited problems.

We are grateful for Professor Lea and her team's thorough and supportive comments on our manuscript.

We have performed polishing and per-particle refinement on our data using Relion3.0 (see for illustration a per-particle FSC estimation below), however this did not lead to any improvement in resolution beyond the second decimal point, nor did it improve the map significantly. We have edited the Materials and Methods accordingly.

In addition, all refinements imposed 5-fold symmetry which could blur out features in the head region. Previous crystallographic structures of pentameric cap heads display significant deviations from true 5-fold symmetry (FliDSe, 5H5T and FlgDSt, 6IEE). While particle numbers may prevent robust reconstruction with C1 symmetry, the issue should be discussed in the manuscript.

We have performed further 3D classification and refinement of our data, using C1 symmetry. As illustrated below, we did not observe major differences within the D2-D3 domains of each protomer:

Class 1:

Class 2:

However, because of the limited resolution of these reconstructions, we cannot conclusively identify subtle changes within the position of individual helices, which we have indicated in the text:

“We also note that in some of the previously-reported FliD D2-D3 crystal structures, there are some subtle differences between subunits of the oligomer. We attempted to determine if this was also the case in our FliDcj map, by performing further refinement without symmetry. The resulting map did not show any significant difference between the subunits (not shown), but the resulting resolution was too low to identify minor conformational changes. Further characterization, using larger datasets and classification, would be required to determine differences in the pentameric subunits of FliDcj..”

In an attempt to circumvent this limitation, we have collected a new dataset for FliD, that contained ~ 10x more particles than the dataset reported in our manuscript. However, we were not able to refine the structure to an improved resolution using this data, and our best reconstruction is stalling at ~ 6 Å, as illustrated below. Potentially many more rounds of classification would be needed to separate the high quality particles and/or identify multiple conformations. However, this will require substantially more work, and we therefore decided to not include this new data in the revised manuscript.

Given the current modest resolution, key features that are later used to construct the functional model need rather better justification in terms of showing the experimental volume. Specifically Local Resolution maps should be shown so that the resolution of the map in the critical termini can be more easily appreciated.

We have included local resolution maps for full decamer FliDcj map, as well as the focused refinement map on the head pentamer domains of FliDcj, in supplementary figure 3. The density in the D0 domains, including the N-terminal stretch, is ~ 4.3 Å resolution, while the rest of the map has a local resolution in the 5-6 Å range, but could be easily modelled based on published crystal structures of orthologues. We have clarified this in the Methods section:

“The leg domains were visibly at a higher resolution than the head domains, therefore a mask centering on the head domain was used for further refinement with C5 symmetry, leading to a map of the head domain to 5.0 Å resolution. Further 3D classification of the masked head domain was used to identify 4 different conformations of the D4 domain not resolved in the full map. The processing pipeline is detailed in Figure S7. Local resolution maps were generated using Relion 3.1 Local resolution function (S3).”

and in text:

“We note however that the map shows a wide range of local resolution, with the leg domain well defined and with visible density for the large side-chains, while the head domain is more poorly defined (Figure S3a, S3b). This could suggest that there is some flexibility in the head domains, or that each subunit adopt slightly different conformations, which were averaged out in our map because of the applied 5-fold symmetry.”

From the figures shown (both main text and supplement) it is clear that there is sufficient resolution to accurately place complete models, but none of the volumes shown give clear support for resolution of side chain orientations. In fact, many of the figures shown in the supplement show side-chain like densities that are unoccupied and bulky side chains lying out of density. If the authors wish to use specific side chain orientations/interactions in generating their models they should support the location of those side chains by demonstrating that the density resolves their location. Even in the helical portions of the N- and C- termini it is unclear where there is sufficient resolution to accurately determine the helical register and the *methods* do not describe how the long helices were built. In light of this panels with the detailed side chain interactions shown in Figure 3a need much stronger support from the experimental data or co-evolution analysis.

Professor Lea correctly indicates that at this resolution, we cannot determine the position of side-chains on figure 3a and we have therefore removed them from this figure.

The model for the D0 domain was built manually coot, and refined in Rosetta and Phenix. For this domain, density for bulky amino-acids is clearly visible, as illustrated on figure S4a of the revised manuscript, which allowed us to determine the register of the helices unambiguously. For the D1, D2 and D3 domains, the local resolution is significantly lower (see comment above), but we were able to build models of these by homology modelling, based on the published crystal structures of FliD orthologues. We have further detailed our model building protocol (see above).

We have also performed co-evolution analysis, based on 283 FliD sequences, using the GREMLIN server. As shown below, we did obtain some co-evolving residue pairs, that support

our atomic model. However none of the high-scoring pairs correspond to the interfaces that we sought to characterize with the mutagenesis study, and we therefore chose to not include this data in our manuscript.

Residue 1	Residue 2	Scaled score	Probability
396_F	401_F	2.455	0.96
107_Q	261_E	2.361	0.94
305_T	402_E	2.330	0.94
271_R	278_D	2.291	0.93
280_G	283_M	2.253	0.92
607_Y	611_A	2.245	0.92
113_V	265_D	2.173	0.90
397_D	400_K	2.148	0.90
159_A	174_I	2.111	0.89
37_T	596_K	2.105	0.88
16_L	19_D	2.071	0.87
259_N	270_F	1.974	0.84
401_F	405_V	1.964	0.83
25_K	606_R	1.840	0.77
29_Q	603_I	1.835	0.77
104_N	263_T	1.816	0.76
262_F	276_V	1.785	0.74
20_T	113_V	1.764	0.73
283_M	295_F	1.759	0.73
154_T	158_L	1.737	0.71
10_G	14_G	1.714	0.70
87_P	291_G	1.713	0.70
8_S	12_G	1.707	0.70

The placing of the N- and C-terminal helices with respect to the helical axis of the filament (N-in) is interesting, as all high resolution flagellar and needle structures have so far shown conservation of assembly with the C-terminal helix forming the inner surface of the filament. While cap structures may be assumed to assemble via a different mechanism, especially as the hook cap FlgD only has an N-terminal helix, the observation of the C-terminal helix on the outside is still noteworthy. However, particularly in light of the fact that early, low resolution models of the type three needle (our own included) interpreted low resolution density as N-

terminus in and were only corrected a much higher resolution, the volume justifying this placing of the helices should be shown and this feature of the structure discussed.

As indicated above, the map is substantially better resolved in the D0 domain, and there is no ambiguity over the position of these helices, as shown in figures S2D and S4 of the revised manuscript.

The comment on the comparison with the filament (flagellum and T3SS) architecture is indeed important, and we have added some elements in the discussion comparing the position of the D0 helices in FliD, compared to the helices of the flagellin and T3SS needle:

“Intriguingly, we note that according to this model, the position of the two terminal helices in the D0 domain of FliDcj is distinct from that of the flagellin (and the related T3SS needle): While in the flagellin the C-terminus helix is facing inwards towards the lumen of the filament, in our structure of FliDcj it is facing outwards. This unexpected architecture is consistent with our model of the tip-filament complex, and supported by our mutagenesis analysis.”

The section comparing homologues has a number of problems.

The negative stain EM data would benefit from a reference structure to orient the reader, either in the form of equivalent 2D from the FliDCj structure or a scaled model. This would hopefully make it clearer that the much larger 6-fold class in the FliDPa and the large 4-fold classes in the FliDSm are likely contaminants.

A 2D class from negative stain micrographs of FliDcj, with the same parameters as the other orthologues, was added to figure 2c.

The text comparing crystal structures and Fig S4b seems to contain 'wrong species descriptions. The text on line 174 refers to an E. coli orthologue pentamer, but this should be Salmonella. Similarly, the legend to Fig S4b refers to FliDSm, when the overlay is with FliDSe (5H5T) and refers to D1 domain overlays when only D2-D3 domains are shown. Clarification should also be provided for which subunit of the FliDSe pentamer was used for the overlay in Fig S4b. 5H5T shows significant deviations in angles between the 5 copies are therefore the choice of chain would impact the angles seen in the overlay of the whole objects.

We are grateful to professor Lea for identifying these typos, which we have corrected. Chain A of the FliDse crystal structure was used for alignment with FliDcj, and we have clarified this in the figure legend. As mentioned above, classification without symmetry did not reveal major differences between subunits, but a larger dataset will be required to identify more subtle differences, however this has not been successful so far.

The resolution of the filament structure is surprisingly poor for a filamentous structure with this number of boxed regions. Have the authors performed any 3D classification or other analysis that might help explain this? Why is the FSC curve so structured?

We have included a data processing scheme for the filament (Supplementary figure 11). We did not pursue further classification, due to our dataset being limited by the usage of a 200kV

inhouse microscope, and the limiting size of the dataset (100 micrographs). The number of particles is somewhat artificially large because we used the rise of the filament symmetry for extracting particles, instead of a full flagellin subunit. therefore each “particle” does not includes a complete flagellin molecule.

Since this, we have obtained a larger dataset, collected on a 300 kV microscope, which we are still processing, and in which we will include further classification to identify different conformations. However this will form the basis of a different manuscript, and is beyond the scope of this study.

The FSC curve indeed is not as smooth as expected. While the reason for this is not entirely clear, we postulate that it might be due to the flagellum consisting of layers with differing local resolutions, thus leading to various drops in the FSC curve at several frequencies. This is supported by the observation that when a mask was used to exclude the outer layer, the resolution was improved, and resulted in a much healthier-looking FSC curve. We chose to show the complete map in figure 4b, as the 11-stranded features are more obvious, but we have included these comments in figure S11, and in the materials and methods section:

“As the FSC curve was not as smooth as expected, a mask was used to exclude the D2 and D3 domain density, and the resulting map of D0 and D1 domains was at a higher resolution with a less structured FSC curve. The processing pipeline is detailed in Figure S11.”

Motility assay and point mutations.

The description in the main text suggests that the effect of the point mutations is to make the filaments brittle. These referees can see no mechanism by which mutations in the cap can lead to assembly of filaments that more readily snap at the cell body, i.e. the opposite end to the cap structure, and it is perhaps surprising that mutations within the cap lead to either no filament or an entirely normal length filament on the bacterium. What molecular events lead to these effects? Are the filaments in the supernatant truly broken off or are have they been nucleated within the supernatant by cap structures that have fallen off the filament?

By brittle, we mean that the flagella were not necessarily cleaved off at the cell body, but rather we observed *C.jejuni* cells with flagellar stubs of varying lengths, with a flagellar fragment near the cells. We have illustrated this in Supplementary figure 9c. Because of this, we do not think that these are spontaneously-assembled filaments. We have however no direct evidence for this, and we have added this clarification in the text:

“All of the mutations still led to bacteria with assembled filaments, of length similar to that of WT bacteria (Figure S9b), demonstrating that the corresponding FliD proteins are still able to promote filament elongation. However, we noted that the filaments are less stable in the mutants, with between 60 and 80% of filaments found broken off at varying lengths from the cell body, versus ~ 20% in WT bacteria (Figure 3c, S9c).”

We do not yet know how FliD mutations cause this, but we hypothesize that it may occur due to incomplete or slightly misplaced packing of the flagellin subunits within the filament, causing the filaments to be are robust and snap off at random points along their length. This is however purely hypothetical, and will require additional characterization beyond the scope of this work, so we did not elaborate futher on this.

Figure S5c should be brought into the main manuscript.

We have moved this panel to Figure 3c

Since none of the point mutants are as fatal in effect as the knockout, have the authors made any double mutants that are?

We have attempted to make double mutants in *C.jejuni*, however we have had technical difficulties, and unfortunately we are no longer able to complete this experiments due to the current shut-down.

The statement that the mutations targeting the proposed filament assembly site are less impactful is not supported by Fig 3b. I620 and L624 are just as important as any of the proposed N-terminus interacting residues. Care needs to be taken in carving up these proposed binding sites. Y617 is only 1 turn of helix up from L620 and has a very similar mutagenesis profile, so categorically attributing the effect of the mutations to one or other interaction is difficult, especially as the proposed flagellar filament interactions can't be a particularly good mimic of the native binding site (see next point). With this in mind, inspection of figure 4d shows that in the constructed model of the cap on the filament, all of the cap helices lie within the filament ones. However inspection of Fig 1a shows that the interactions that build the decamer are side-ways interactions between helices at the same radius and it is therefore unclear how the interactions that build the decamer are a proxy for the any of the interactions that lead to assembly on the filament. The fact that all the interactions will be different is also not acknowledged in discussion in the text.

We certainly agree that mutants on residues in both interfaces significantly affect motility, and we have revised the text accordingly:

“Mutating these residues impacted motility in a similar manner to that of the residues of the pentameric interface (see above), confirming that the hydrophobicity of this interface also contributes to FliD function in filament assembly.”

As indicated above, we are confident in the register of the two helices, and therefore we clearly have two distinct interfaces, at least in our FliD decameric structure. However, the residues chosen for mutation were done so not only due to their localisation but also their conservation across bacterial species (see figure 2d). We therefore think it is significant that some mutations in the second interface (L628, I635) do not majorly affect motility. How this translates to the actual FliD-filament interaction is indeed based on our model, and may in reality significantly differ from the pentamer-pentamer interface. We have included these points in our discussion section:

“We note however that because of the symmetry mismatch, a different set of interactions are formed on each side of the cap complex. It is possible that this is achieved because the interface is largely composed of non-specific hydrophobic interactions. Alternatively, there could be some structural changes of the D0 domain of FliD that our structure doesn't capture.”

Proposed mechanism – there were various aspects of the proposed mechanism that these referees

feel either need further explanation, further experimental support or perhaps simply less absolute language in their description.

Is the closed pentameric complex the physiologically relevant state? Whilst the idea that the object seen in isolation is the one that will be found at the tip of the growing filament is attractive, the authors should be careful in how firmly this is stated. The mutations do not prove the physiological relevance of the complex, only that the termini are important for function. These referees feel that unless the cap can be visualised in situ, or the hypothesis supported in another way such as by construction of fully functional disulphide locked cap complexes, then the language used should be moderated.

We absolutely agree that some aspects of the proposed mechanism are highly speculative at this point. We have moderated our language, and added further references and explanations, in the discussion:

“Prior to assembling the filament, FliD is exported through the lumen of the flagellar hook, and assembles at the end of the hook-filament junction. It is not known whether individual FliD molecules attach (potentially with a helical architecture), or whether it forms a pentamer first, which then docks onto the hook-filament junction..... In order to accommodate the next flagellin subunit, conformational changes need to occur to open an adjacent binding pocket. We propose that the folding of the new flagellin protomer leads to dislodging of the cap complex, that rotates by $\sim 65^\circ$ (4), thus positioning an adjacent cavity of the cap complex close to the next flagellin insertion site (Figure 5).. This hypothesis agrees with previously-proposed mechanisms of flagellar elongation.”

The authors do not really discuss how, during assembly on the filament, the cap complex would form this planar complex rather than a helical one. The cap must initially assemble at the end of the hook filament junction which shows helical symmetry, and the initial binding event must be tight enough to allow capture of the cap monomers being secreted. The cap may initially therefore follow the helical symmetry and then rearrange into the planar structure observed here. One assumption would presumably be that this object is much more stable than 5-copies in a helical arrangement? For all other stable flagellar/type 3 complexes *co-variation of residue pairs* has been able to provide strong support for threading of sequence in modest resolution volumes. Can the authors provide such data to support construction of their model?

The idea of the helical complex being formed first and then its rearrangement is absolutely of primary interest, but our data does not allow us to answer this. We have however included this in the discussion:

“Prior to assembling the filament, FliD is exported through the lumen of the flagellar hook, and assembles at the end of the hook-filament junction. It is not known whether individual FliD molecules attach (potentially with a helical architecture), or whether it forms a pentamer first, which then docks onto the hook-filament junction.”

As suggested, co-evolution analysis was conducted between FliD and flagellin sequences, and included in the manuscript (Table S3). This data largely support our FliD-filament model, with many residues from the FliDcj -terminus showing significant co-evolution with flagellin N-

terminal region. The residues that were chosen for mutation in the C-terminal interface we proposed are all present above, mostly interacting with hydrophobic or uncharged side chains.

How was a completely symmetric, planar pentamer docked onto the helical filament, i.e. how were decisions about rotational position and height within the assembly made? Surely one problem with this symmetry mismatch is that every FliC/FliD interaction would be unique. The authors then state that “Because of the symmetry mismatch, this interaction is not present on one side of the cap complex”. Presumably this is reflected by the arrow in Fig 4d (although this is not referenced in the figure legend), but there isn’t a clear cavity in the figures shown and surely any such cavity would always be smaller in this model than in a model that allowed flexibility of the legs.

We have further expanded our description of the filament-FliD model, which was based on the low-resolution tomography map of FliD bound to a hook in FlaB knockout mutant of *Borrelia burgdorferi* (Zhang K. et al., 2019), including supplementary figure 12, and new section of the Materials and Methods:

“The low-resolution tomography map of FliD bound to a hook in FlaB knockout mutant of Borrelia burgdorferi was used as a volume for docking atomic models. The pentamer FliDcj model (this study) was used for the region corresponding to the cap, and the P.aeruginosa flagellum model (PDB: 5WK6) was used for the filament, as our filament map was not high enough resolution to obtain an atomic model. The FliDcj pentamer could be fitted in three positions, as shown in Supplementary figure 12. The central position is clearly the model with least steric clashes between FliD and flagellin, and we therefore used this for the mechanistic model shown in Figure 5. We note that there is some extra density of the tomography map above the density attributed to D2-D3, which could correspond to the D3 head domain present in Borrelia burgdorferi FliD (not shown).”

The final model invokes rotation of a rigid cap pentamer but various elements of the proposed mechanism need further explanation, including details such as why folding of flagellin leads to a 35° rotation of the cap (360/11≠35 nor is 360/5)? The major difference between this model and earlier ones (which also invoked rotation) is the idea that the cap legs are locked by interactions within the cap complex and that their flexibility is therefore not a key element of filament assembly. As noted above, further evidence either in the form of strong *co-variation* of the residues involved in assembling the cap complex or by demonstrating that covalently cross-linking the complex does not perturb function would be required to firmly refute any role for flexibility.

We are pursuing structural characterization of the filament-cap complex, including tomography, cross-linking, single-molecules approaches etc... but this work will be forming the basis of an additional publication, and is beyond the scope of this manuscript.

It is true that our data does not preclude movements and/or asymmetry in the leg domains within the native complex, however there is also nothing in our data that suggest that there is any, and as we suggest in the discussion section a model without such motion could be envisioned.

“We note that previous studies, based on low-resolution tomography data, have suggested that the D0 domain of FliD might be dynamic, with the leg domains opening and closing to promote filament elongation. In contrast, in our structure of the cap complex, the D0 domain is rigid, locked in position by the N-terminal stretch. This structure therefore could provide evidence for a different mechanism, which had been proposed previously, whereby the cap complex acts as a rigid cog that rotates during flagellum elongation. Nonetheless, further experiments to characterize the flagellum-cap complex at high resolution will be required to confirm this model, and in particular to observe if the rigidity of the D0 domain, as observed in our structure of the cap complex, is also present when FliD is bound to the filament.”

We also thank professor Lea for identifying an error on our part: we speculate that the cap would need to do 2×360 degree rotations to incorporate 11 subunits with its 5-fold symmetry, and therefore the angle of rotation of a flexible or rigid structure would be $720/11 = \sim 65$ degree rotation. We have changed this in the text.

Reviewer #2:

The authors present the first sub-nanometer structure of the in-vitro purified cap complex from *Campylobacter jejuni*. The main finding is that it has 5-fold symmetry together with some evidence that the identified interactions are important for motility. Furthermore, they have reconstructed the native filament and show that it consists of 11 protofilaments, which corrects previous claims by another group. Despite the limited resolution, they were able to build an atomic model and propose a hypothesis for filament-cap interaction and elongation.

The manuscript is well written and generally laid out clearly. However, the author's conclusions are often somewhat far-reaching, which may not be justified by the presented evidence. I will provide detailed examples below. The discussion is unusually short and not well separated from the narrative in the results section.

The presented results are overall novel and of good quality.

The finding that the filament is 11-stranded is a major result on its own and became an integral part of their interaction model, yet it is not described in great detail. The elongation model is very speculative and rests mainly on the author's imagination. The results section should be simplified to contain the findings without adornment – insights and conclusions should be moved to the discussion, which should be expanded. The reality is that the resolution of the structures presented in this study is at the limit for making functional claims at the molecular level. Nevertheless, the main findings represent an advancement and can stand on their own.

Detailed comments.

42 apparatuses

Like 'status', the Latin plural of the noun 'apparatus' is 'apparatus' – I would refrain from anglicizing it and replace it by "molecular machinery" or "mechanisms", etc.

We acknowledge this reviewer for his/her positive comments on our manuscripts. We have replaced the term "apparatus" with "*molecular machinery*", as suggested.

79 "we show that these interactions are essential for cell motility"

The presented evidence (Fig 3b) indicates that mutation of these residues affects motility. However, they are not essential, since these mutant strains are still motile.

We have changed this sentence to "*We show that disrupting these interactions have a significant effect on cell motility*"

82 "the pentameric state of FliD within the cap complex is likely universal"

The term "universal" is not justified based on a sample size of 3 species. Should provide broader evidence if this statement is kept – e.g. sequence alignments of the major bacteria within the kingdom. Or tone down the phrase to something like "...may be a common motif within the kingdom of bacteria"

We have changed this sentence as suggested.

105 While the FSC threshold can be cited to arbitrary precision, $1/100\text{\AA}$ is nonsense. Change to 4.7\AA .

We have removed the 2nd decimal.

113 “suggests that complex is dynamic, which a hinge between leg and head domains”
The lower resolution in the head domain could be a result of alignment errors or the averaging of multiple still unseparated classes and does not necessarily suggest a dynamic structure. What is the character of this hinge? Where is it located? How did the authors arrive at this conclusion?

We have replaced this sentence with “*This could suggest that there is some flexibility in the head domains, or that each subunit adopt slightly different conformations, which were averaged out in our map because of the applied 5-fold symmetry.*”

118 “build the atomic model for the leg domain de novo”
No validation of the new model besides Ramachandran statistics is provided. I suggest calculating a Fourier shell correlation between map and the synthetic model-derived density and add this cross-correlation curve to Fig. S6b. That curve would reveal up to which resolution the model detail is consistent with the experimental map. Any interpretation of molecular details beyond this resolution is not justified.

We have added the map-to-model FSC curves to the Supplementary figure 4, along with the localized curves for each domain.

130 “N-terminal residues are extended into a stretch that folds...”
at 4.7\AA resolution, the tracing of a single polypeptide chain is expected to be very difficult. Please provide examples of this map feature together with the fitted model. What were the B-factors of the refined amino acid residues in this region?

As indicated in our response to Professor Lea above, and as illustrated in the local resolution map shown in Supplementary figure 4, map for the N-terminal stretch is closer to $\sim 4.3\text{\AA}$ resolution, which allowed us to confidently build the backbone. Rosetta modelling was further used to validate the sequence assignment for these residues, and Phenix was used for refinement, including B-factors. This has been clarified in the Materials and Methods section:

“For the D1-D3 domains, a homology model was generated with PHYRE2, using the FliDec crystal structure (PDB:5H5V) as a template. Domains D2 and D3 were fitted into the sharpened head domain map in Chimera. Domain D1 was fitted into the sharpened full FliDcj map in Chimera. Domain D0 was built into the sharpened full FliDcj map using Coot.”

The B factor for each of the residues of the N-terminal stretch is listed below:

MET 1	86.25
ALA 2	62.49
PHE 3	67.33
GLY 4	71.78
SER 5	53.49
LEU 6	51.12
SER 7	71.75
SER 8	63.39
LEU 9	56.32
GLY 10	43.51
PHE 11	30.85
GLY 12	20.32
SER 13	18.75
GLY 14	23.11
VAL 15	47.19
LEU 16	68.27
THR 17	61.98

144 “low res = flexible”

Again, this is not a universal truth. From the sharpened maps in Fig.S3b it appears that the map features in that region are below nanometer resolution. How many particles were included in the sub-classification shown in Fig.S3c? At such low resolution and presumably after low-pass filtering, the iso-contour of the envelope of domain D4 is not indicative of distinct conformations. I suggest omitting this conclusion about flexibility altogether and leave it at the fact that additional density corresponding to D4 was observed.

We have added a data processing scheme for the FliD data in Supplementary figure 7, that includes the various classification and particle numbers. It is true that at this resolution, the structure of the D4 domain is not clear, however it is very apparent that its position relative to the other domains differs between the classes. We have therefore removed the term “flexibility” in the text, and instead mention that different classes corresponding to distinct relative positions of D4, were observed:

“Further 3D classification without imposing symmetry revealed at least 4 potential distinct positions for this domain (Figure S6b), but the resolution obtained was limited by the number of particles used for reconstruction.”

152 “first high resolution structure” of an “intact FliD protein”

Since the “resolution revolution”, 4.7Å is not “high res” anymore, sorry. If superlatives are needed, replace with “our structure, at 4.7 Å, represents the highest resolution of this complex to date”, or similar. The 4.7 Å structure is furthermore the E.Coli-expressed, in-vitro purified cap dimer. I doubt this qualifies as “intact”, whatever this means. Replace e.g. with “full length”, if the case.

156 RSMD

We have edited the manuscript accordingly

161 “dramatically different ... D1 position ... suggests that the hinge between D1 and D2 is flexible”

These are two different classes within the phylum of proteobacteria (gamma and epsilon)! This conclusion and the site of the hinge region are speculative at best.

Our hypothesis that the hinge region is flexible is indeed speculative, and would require further structural/biophysical evidence. This has been changed in text:

“This could occur because the hinge between D1 and D2 is flexible. Alternatively, it could be due to the fact that in FliD_{cj} the D4 domain protrudes from the D1 helical bundle and thus a more planar conformation, as observed in FliD_{se}, may be sterically clashing with D2-D3 in the case of C.jejuni FliD (Figure S8b), leading to a conformation distinct to that of other species.”

175 “large degree of plasticity between D2-D3”

Again, while this observation is interesting, it is not a direct observation within the FliD(cj) sample, to which the statement applies. The only conclusion that can be drawn from this comparison is that the two structures are different. Such speculations should go to the discussion section, not in results.

We have moved this to the discussion.

Fig.2c no scale bar

Important, since the legend talks about size comparison. Please provide absolute dimensions.

We have included scale bars as suggested

207 “we also note that N- and C- termini of FliD are highly conserved across species...”

Which species? The examples given are from different classes. Across phyla? Within the domain? Provide sequence alignment or reference to support this statement.

The alignment for FliD was amended to include sequences from across the Proteobacteria phylum, as well as with *B. subtilis* (a gram-positive bacterium), and moved to the main figure (Figure 2d). The text was also changed to reflect this:

“We also note that both the N- and C- termini of FliD (residues 1-46 and 577-643, respectively, in the FliD_{cj} sequence) are conserved across a wide range of bacterial families within the Proteobacteria phylum as shown in the multiple sequence alignment (Figure 2d).”

Fig.3a need to show a superposition of the electron potential map to support the fitting of these side chain rotamers.

As mentioned in the response to Professor Lea (see above), the resolution of the map is not high enough for us to confidently identify the side-chain rotamers. The model was refined through multiple rounds of refinement in Rosetta and Phenix, and we are confident that they correspond to an energy minimum, but they are indeed not based on the data. We have therefore removed the side-chains from Figure 3a.

Fig.3b What was the number of times the motility tests were repeated for each of these mutations? There are error bars for the standard deviations, but N is not provided in the legend nor in the methods section.

Each sample was measured in triplicate per plate and the average value was used in 3 independent experiments to produce a value of N=3. This was clarified in the figure legend and materials and methods section.

219 “this confirms that the hydrophobic properties are critical for motility”

While this is an interesting experiment, this conclusion is not valid as stated. Hydrophobic residues were mutated to A,L,S and 20-50% reduction in motility was detected. However, whether this effect is due to the hydrophobic character of the side chains or other reasons (steric effects, a variation in filament length due to altered assembly kinetics, altered filament geometry, etc) is unclear. Furthermore, FliD is not the chief determinant for motility, but it regulates filament assembly, as stated by the authors. The effect should simply be described here. Any interpretations should be toned down and moved to the discussion section.

We have removed this from the results section, and instead mentioned it as part of a longer section on the filament-cap model in the discussion:

“It is possible that this is achieved because the interface is largely composed of non-specific hydrophobic interactions. Alternatively, there could be some structural changes of the D0 domain of FliD that our structure doesn’t capture.”

236 “evidence from tomography indicates that this interaction is not physiological. However, since it is observed both in FliD(cj) and FliD(se) we postulated that it mimics the interaction between FliD and the filament”

Both FliD(cj) and FliD(se) were in-vitro assembled cap dimers, which are not physiological. End of story. Any conclusions in this direction are pure imagination.

We have removed this from the results section.

266 “to avoid biases due to symmetry, we initially performed reconstruction without any helical symmetry applied”

How was the initial reconstruction created? This is not described in the methods section. It is remarkable, that for a straight helical filament such evenly distributed angular sampling could be achieved. The cross section in Fig.6 shows very little anisotropy. What structure was used as initial alignment reference? More details are required.

We have included these details in the Materials and Methods section (see below)

270 “we refined the map further by applying helical symmetry, with 65.4 deg twist and 7.25 Å rise”

How were these initial helical parameters derived? Are these values the refined parameters after helical refinement with reliction? If so, what were the starting values?

The data processing scheme is shown in Supplementary Figure 11. The initial reference model was used from 70 Å low pass filtered map of *P.aeruginosa* flagellar filament EMDB:8855, which was cropped to the appropriate box and pixel size to fit our data. This was done to remove any features from a map and create a smooth cylinder surface to align particles on.

We refined the map further by applying helical symmetry initially with 65.5° twist and 4.7 Å rise as per the values used to reconstruct the filament of *P.aeruginosa* in our reference map (Wang et al., 2017). However, this reconstruction did not converge to a map with more defined features and a smooth FSC curve. Thus we performed multiple 3D refinements with a search range for both the twist and the rise of our filament, which converged on a 65.4° twist and 7.25 Å rise, which allowed us to reach ~ 8.6 Å resolution (Figure S10c). We have clarified this in the Methods section:

“70 Å low pass filtered map of P.aeruginosa flagellar filament (EMDB:8855), in a form of a smooth cylinder with the correct dimensions, was used as a reference for all 3D refinements. Without symmetry, the structure refined to 27.2 Å resolution, but when helical symmetry was applied, the final resolution after further classification and refinement was 8.6 Å, with a 65.4° twist and a 7.25 Å rise. As the FSC curve was not as smooth as expected, a mask was used to exclude the D2 and D3 domain density, and the resulting map of D0 and D1 domains was at a higher resolution with a less structured FSC curve. The processing pipeline is detailed in Figure S11.”

274 “the fact that we can reach only limited res is not surprising, since we have L and R...” Or that there were not enough particle images, or the image quality wasn’t good enough, or alignment and classification were incomplete...” There are many reasons for limited resolution. Mix of conformations is only one of many possibilities.

We have included these elements in the text:

“The fact that we can only reach limited resolution is perhaps not surprising, since we likely have a combination of L and R conformations for the flagellin. Other reasons for low resolution may include the amount and quality of the micrographs (collected on a 200kV microscope), as well as the limited number of extracted particles. Nonetheless, this data conclusively demonstrates that the P.aeruginosa flagellum filament is 11-stranded, and not 7-stranded as reported previously.”

285 and Fig. 4. Why use the *P.aeruginosa* filament and why not show the 8.5 Å C.j. map together with FliD(cj)? Any “atomic” model over-interprets the present data. The new structures in this

paper are nice, although at limited resolution. Simply show them as is.

We used an atomic model instead of the filament map, because the model contains a growing filament end with the filament monomers at different levels, unlike the map which is flat at the ends (box edge). This would allow for a better fit of the FliD_{cap} as we assume that the cap is located at the growing end of the filament.

It would further be prudent here to expand upon the 11-stranded filament structure and describe it in greater detail.

Because of the limited resolution of this reconstruction, we feel that further description of the 11-stranded filament might be too speculative. We are performing further analyses on this, using a larger dataset collected on a more stable microscope, which we hope will allow us to draw more convincing conclusions on this structure. This is however beyond the scope of this publication.

295 “in a range of species”
in 3 species, as described in this paper

303 “hydrophobic residues act as chaperonin-like environment to promote folding”
This idea has been put forward before, as referenced, and should not be presented as a novel hypothesis.

306 “a universal mechanism for cap-mediated filament elongation”
The claim “universal” is based on 3 examples of structures at intermediate resolution, without any additional evidence, not even sequence alignments of the supposedly important hydrophobic residues.

We have modified our manuscript as suggested.

310-315 This is pure imagination of the authors. Please provide evidence for each of these steps or rephrase this discussion paragraph as a literature review with cited references.

We thoroughly revised this section, to include further references, and to highlight the aspects that remain speculative.

“New unfolded flagellin molecules are secreted through the filament, and our model suggests that they would emerge in a chamber inside the cap complex (1). According to our model, there is a gap between the D0-D1 domains of FliD that is not obstructed by the filament on one side. We propose that unfolded flagellin molecules might exit the complex through this cavity (2). While there is currently no direct evidence for this, we note that the residues Y315 and N316, which are facing this cavity, are conserved in proteobacteria (Figure 2d), and across all flagellated bacteria (not shown) . Once outside of FliD, we propose that exposed hydrophobic residues act as a chaperone, and promote flagellin folding in its insertion site, as previously suggested (3) . In order to accommodate the next flagellin subunit, conformational changes need to occur to open an adjacent binding pocket. We propose that the folding of the new flagellin protomer leads to dislodging of the cap complex, that rotates by $\sim 65^\circ$ (4), thus positioning an

adjacent cavity of the cap complex close to the next flagellin insertion site (Figure 5). This hypothesis agrees with previously-proposed mechanisms of flagellar elongation”

316-325 rotation model

“our structure does not support this model, as we show that the N-terminal stretch of FliD is essential for filament elongation and maintains the leg domains in a rigid position”

Where do the authors show that the N-terminal stretch is essential for filament elongation?

The term “rigid” is derived from the fact that the leg domains have locally higher resolution in the FliD dimer. However, this may be a direct result of the dimer interaction and may not be physiological. The same applies to the conformation and interactions of the N-terminal stretch. None of these observations support the conclusion that the function of the N-terminal stretch is to keep the legs in a rigid position – it could as well be interacting with flagellin in the physiological state. Therefore, the conclusion that these data are incompatible with a dynamic leg model is far-fetched.

This reviewer is correct to highlight that our structure does not preclude a dynamic leg model, as highlighted also by Professor Lea. However, our structure does not provide evidence in favour of such model either. Both dynamic and static leg models have been proposed, and our structure provides evidence supporting the static model, but indeed further data will be required to conclusively determine which model is accurate. We have clarified this in the text:

“We note that previous studies, based on low-resolution tomography data, have suggested that the D0 domain of FliD might be dynamic, with the leg domains opening and closing to promote filament elongation. In contrast, in our structure of the cap complex, the D0 domain is rigid, locked in position by the N-terminal stretch. This structure therefore could provide evidence for a different mechanism, which had been proposed previously, whereby the cap complex acts as a rigid cog that rotates during flagellum elongation. Nonetheless, further experiments to characterize the flagellum-cap complex at high resolution will be required to confirm this model, and in particular to observe if the rigidity of the D0 domain, as observed in our structure of the cap complex, is also present when FliD is bound to the filament.”

Reviewer #3:

This straightforward and insightful article describes the author's work to better understand the mechanism of bacterial flagella filament elongation. To do this, they focus on the role of FliD, the "cap" protein believed to ride the tip of the elongating flagellar filament. Using the well-established alternatively flagellar model organism, *Campylobacter jejuni*, the authors purify a protein complex and determine its structure by cryoEM. Furthermore they use their structure to propose mechanistic hypotheses about filament that they subsequently test by mutagenesis and monitoring of swarm plate motility. The author's structure suggest a mechanism for FliD oligomerization and how it chaperones flagellins to polymerize into the flagellar filament.

The field has a number of controversies, centring around the belief that different organisms build flagella with different oligomeric states in different components (for example, it has been suggested the *Campylobacter* filaments are formed of 7 protofilaments in place of the normal 11, and that FliD oligomers are usually pentameric but sometimes tetrameric or hexameric). This study convincingly puts to rest this unlikely suggestion by showing that *Campylobacter* has a flagellar filament with 11 protofilaments and a pentameric FliD cap.

The writing is overall good and clear, although at times word choice is somewhat confusing, and the figures I found to be less illustrative of the important findings than I had hoped. But overall an interesting and important study.

There is one major point that will need addressing before acceptance for publication, however, beginning around line ~226: The authors show that a series of point mutations in the N-terminal portion of FliD still allows for the synthesis of a full-length, WT filament, but that these filaments are more "brittle" than a filament that polymerizes under a WT FliD cap. The evidence for this claim is the increased number of broken filament pieces present in negative-stain micrographs of these FliD mutants relative to WT. The presence of these filament pieces (along with the apparent WT lengths of the FliD-mutant filaments) must mean the defect caused by the FliD mutations is affecting the filament itself (i.e. if the defect were at the FliD/filament interface, as would be expected, one would expect shorter filaments but not a greater number of broken filaments than WT). Thus, the authors seem to be implying that the pentameric FliD cap at the very end of the multi-micron filament is modulating the flexibility/rigidity of the filament (and possibly the hook?) as a whole. If true, this is a novel and, to my knowledge, completely unprecedented role for FliD. If it is true that the filaments are breaking at the rod-hook interface, and not simply at random along the filament, this finding is even more significant (and confusing).

However, the sole piece of evidence for this claim is the observation of more filament pieces on grids by negative stain EM. But, even with WT *C. jejuni* and with very gentle pipetting using cut tips to avoid filament shearing and frothing the cell suspension, there will still be a significant number of filament pieces on the grid, and there is a lot variability day to day, grid to grid and grid-square to grid-square in the number of disattached and broken filaments seen by negative stain TEM (at least in our hands).

The authors will need to provide additional experimental evidence for the increase in brittleness in the FliD N-terminal mutants. This could possibly be done by, e.g., western blotting or Coomassie staining for flagellin in the culture supernatants of the various mutants, among other

methods. It would be welcome if the authors could also speculate a mechanism for how the FliD cap is tuning the rigidity of the entire flagellar filament.

We thank this reviewer for providing very constructive suggestions.

The “brittleness” that the authors are referring to has been clarified in text:

“However, we note that the filaments appear less stable in the mutants, with between 60 and 80% of filaments found broken off at varying lengths from the cell body, versus ~ 20% in WT bacteria (Figure 3c, S9c). Further characterization will be required to confirm if this is indeed the case, and other effects (such as kinetics of assembly) could also result in similar phenotypes.”

and in Supplementary figure 9c (See response to Professor Lea’s comment above).

We acknowledge that further work would be required to characterize this, but these are not trivial, and beyond the scope of this study. For example, the suggested experiment to quantify the flagellin content from the supernatant is very challenging, because in liquid media the *81116 C. jejuni* strain does not grow flagella, and therefore all of our experiments need to be conducted on plate-grown cultures. In any case, the central point that we are making is that while our mutants significantly reduce motility, they are still able to assemble full flagellar filaments.

About the rigidity tuning mechanism, our current working hypothesis is that a FliD complex where interaction with the filament is altered (as is the case with our various mutants), might cause nascent flagellins to misfold or misposition, which will cause brittleness. We therefore do not think that there is a “long-range” effect with our FliD mutants, but rather that the assembly process is somewhat less efficient, leading to slightly misassembled filaments throughout. The breaking events are stochastic, and can occur at any points. This is however completely speculative, and will require a lot further characterization, probably in a different model organism for the reason mentioned above, and therefore beyond the scope of this work.

To answer the reviewer’s point on handling, we used the same method and batch of cells/grids/stain for analysing the wild type and the point mutant strains via EM. We proposed said “brittleness” due to our observation of a large difference in sheared or broken off flagella between the WT and the mutant strains which were subjected to almost identical handling.

Additionally, the authors state that 60-80% of filaments in the mutant backgrounds are disattached vs. 20% for WT. This needs clarification. What exactly are the authors quantifying and how are they doing it?

The WT, knockout, complement and all point mutant strains were handled in parallel. Flagellum attachment was determined through imaging cell cultures by TEM. Only filament with 2 or more inflection points were included, to avoid counting various fragments from the same filament. The percentage of attachment was calculated as a proportion of the total flagella observed per mutant. The ratio of flagella attached and flagella unattached in respect to the total flagella number

counted was converted into percentages and plotted. Example of attached and broken filaments are shown in figure S9c

Does this mean that 60-80% of cells have an entire filament (with hook?) laying directly next to the pole of a non-flagellated cell, and that this filament unambiguously snapped off that specific cell? If so, how do the authors know it came from that cell? How do the authors know that an aflagellate cell had a filament break off and simply hasn't constructed a filament in the first place?

This reviewer is correct to highlight that our data does not prove that that particular flagellum came from an adjacent cell or if a cell did not construct a filament. However, as we have plotted the flagella attachment as a percentage of total flagella observed, this gives us a rough idea whether statistically the flagella are more likely to be attached or unattached to a cell. We did not look attachment at a per-cell basis but as a total of all flagella observed in the sample.

Are the filaments disattaching at the hook-rod interface?

We always see a small stub indicating that the hook is still attached to the cell. However, the length of the stub differs between cells. Once the stub has 2 or more inflections we consider it a flagellum and it is counted as attached. This is apparent in the example micrographs shown in Figure S9c of the revised manuscript.

How do they know whether a cell with a stubby filament has a broken filament vs. being in the middle of constructing a new, unbroken filament?

We acknowledge that we cannot conclusively demonstrate this, and further experiments would be required for this. However, in many instances, we see the broken filament next to the cell in negative-stain (see supplementary figure 12), which to us suggests that they are likely broken. Further work would be required to further characterize this, but it is beyond the scope of this study.

Finally, from the micrographs provided, it appears to me that some of the FliD mutants (e.g. L628A, I620S, L9S) ARE making shorter filaments than WT. Have the authors attempted to measure filament lengths of various mutants? Even if the difference in length were 10-20%, this would be significant.

We have measured the filament length in the various mutants using ImageJ software and tracing the filaments, but we did not see statistically-significant difference in length, so we did not include this data in the manuscript. This data is shown below:

The color coding is as in Figure 3b in the manuscript.

I also have a number of minor suggestions:

Line 22: use of the word “allow” is a strange choice, and may be directly misleading to newcomers to the field. “Allow” suggests that bacterial flagella relieve some unstated inhibition of motility. Suggest a word like “drive” or “facilitate”.

Line 31: the 11-protofilament flagellar filament is important, and my feeling is that the authors could claim more credit for bringing this to light.

Line 69: “often the most common case” is strange wording. “the most common case” is singular, so how can this be “often”? Suggest removing “often”.

Line 71: suggests two polar flagella, two at each pole. Suggest clarifying with “one each at each pole” or similar.

We have modified the manuscript as suggested

Figure 1: I was somewhat disappointed not to be helped to understand this figure better, which looks very nice, but could communicate keys findings more clearly. I’d suggest a topology map of the protein fold would be a helpful addition.

The topology map was too large to be included in the main figure, but it is included in the revised manuscript in supplementary figure 5.

Panel A would benefit from labeling (with labeling lines) the four domains in one of the monomers). Panel B would benefit from colour-coding by sequence position from N- to C-terminus instead of by domain: the domain labelling already present is more than sufficient, and one of the interesting features of the fold is that it folds back upon itself, with D0, D1, etc composed of sequence-separate segments. Finally I would like to see the position of D4 labelled

explicitly in the models, especially panel C (as in, e.g., Fig S3). At the moment the molecular model suggests, misleadingly, that D4 does not exist.

We have modified the figure as suggested. However, to retain the continuity with the alignment and further colour scheme of the domains, it was better to leave panel B coloured by domain rather than by sequence position. With a topology map and sequence alignment in the supplementary, we hope that it will be clear enough. In panel C we did not add a circle indicating the missing D4 domain, as the figure would be too crowded. Instead, we have added a note in the figure legend to indicate that D4 was missing.

Line 115: I'd like to see more description of how the authors combined these two partial maps so that they could build an atomic model.

We have clarified this in Materials and Methods as follows:

“For the D1-D3 domains, a homology model was generated with PHYRE2⁴⁰, using the FliD_{ec} crystal structure (PDB:5H5V) as a template. Domains D2 and D3 were fitted into the sharpened head domain map in Chimera. Domain D1 was fitted into the sharpened full FliDcj map in Chimera. Domain D0 was built into the sharpened full FliDcj map using Coot.”

Paragraph of line 121: I think the readers deserve some help here in outlining that “D0” and “D1” are structurally similar between flagellin and FliD, and that this likely reflects that they are homologous, and that “D0” in flagellin is synonymous with “D0” in FliD. Don't make them have to figure it out for themselves.

We have modified the text as suggested

Line 131: “interacting with the preceding subunit” really deserves a dedicated panel in Figure 1 to illustrate this.

Because of size limitations, it was difficult to include this in figure 1. This interaction is shown in figure 3a, and to a large extent in figure S12b of the revised manuscript, which we hope will be sufficient to illustrate the details of the interface.

Line 156: “RMSD”

Line 181: “verify”: if the authors set out to “verify” their hypothesis, this suggests to me that they were “trying” to confirm their hypothesis. A more satisfying word would be “test”.

Line 188: what percentage? Needs to be directly stated in Results.

Line 189: what percentage? Needs to be directly stated in Results.

Line 190: “significantly”: by what statistical test? By how much? I suspect the authors mean to say “substantially”.

Line 217: “reduced motility”: I had to search in the figure legend to find, specifically, what the authors mean by “motility”. They mean swarm diameter on swarm agar, which does not directly reflect motility—it might reflect chemotaxis. Suggest using more specific language so as not to mislead the reader. (though I have no doubt that, in this case, it reflects motility).

Line 253: “homologues” is more accurately “paralogues” (I’d congratulate the authors on correct use of the term “ortholog(ue)” throughout!)

We have modified the text accordingly

Paragraph of line 264: did the authors try imaging a straight mutant, or the G508A mutant? I’m not asking that they do this if they haven’t already, but if they did, I’d be very curious.

We have not attempted this mutation.

Figure 4a: could have close-ups of filament

We have modified Fig 4a accordingly

Fig 4D, 5: the mixture of density map and cartoon are confusing and don’t help the reader understand the model. Suggest standardizing on one or the other?

We have moved the FliD-filament model to the Figure S12, and further expanded on it. These include atomic-model representations, which should help with clarity.

REVIEWERS' COMMENTS:

Reviewer #1 (Remarks to the Author):

The current version addresses the majority of our points and now seems suitable for publication.

A couple of minor things that might help

1. Figure 1 - despite the authors' assurances, these referees really can't see the relationship between the colours in panel a and in panel c
2. The new supplementary figure 4 has no numbers to orient the reader on panels a and b - if any of the N to C termini co-varying residues do define the relative heights of these helices it would be helpful to show these as the density (at least in a 2D figure) is still less than compelling as way to uniquely determine the register of these helices.

Reviewer #3 (Remarks to the Author):

The authors have addressed out concerns.

Reviewer #1:

The current version addresses the majority of our points and now seems suitable for publication.

A couple of minor things that might help

1. Figure 1 - despite the authors' assurances, these referees really can't see the relationship between the colours in panel a and in panel c

We acknowledge Professor Lea for noticing that some discrepancies remained in the color scheme in Figure 1, between panel a and panel c. We have now rectified this, and the color scheme is now matching.

2. The new supplementary figure 4 has no numbers to orient the reader on panels a and b - if any of the N to C termini co-varying residues do define the relative heights of these helices it would be helpful to show these as the density (at least in a 2D figure) is still less than compelling as way to uniquely determine the register of these helices.

To facilitate the orientation of fig S4, we have added numbering of the first and last residues, for each of the domains shown. In addition, in panel a, we have labelled some bulky residues where side-chain density was clearly visible, which helped with registry of the helices. Those residues for which co-variation data support the modelling are colored in red.

Reviewer #3:

The authors have addressed out concerns.